# A satellite view of the exceptionally warm summer of 2022 over Europe

João P. A. Martins[1,2], Sara Caetano[1,3], Carlos Pereira[1], Emanuel Dutra[1,4] and Rita M. Cardoso[4]

[1]Instituto Português do Mar e da Atmosfera, Lisbon, 1749-049, Portugal
[2]European Centre for Medium-Range Weather Forecasts, Robert-Schuman-Platz 3, 53175 Bonn, Germany
[3]Direção-Geral do Território, Lisbon, 1099-052, Portugal
[4]Instituto Dom Luiz, Faculdade de Ciências da Universidade de Lisboa, 1749-016 Lisboa, Portugal

*Correspondence to*: João P. A. Martins (joao.martins@ecmwf.int)

**Abstract.** Summer heatwaves are becoming increasingly dangerous over Europe, and their close monitoring is essential for human activities. Typically, they are monitored using 2 m temperature from meteorological weather stations or reanalysis datasets. In this study, the 2022 extremely warm summer over Europe is analyzed using satellite land surface temperature (LST), specifically the LSA-SAF All-Sky LST product (available from 2004 onwards). Since climate applications of LST are still poorly explored, heatwave diagnostics derived from satellite observations are compared with those derived using ERA5/ERA5-Land reanalysis data. Results highlight the exceptionality of 2022 in different metrics such as mean LST anomaly, area under extreme heat conditions, number of hot days and the Heatwave Magnitude Index. In all metrics, 2022 ranked first when compared with the remaining years. Compared to 2018 (next in all rankings), 2022 exceeded its LST anomaly by 0.7 °C and each pixel had on average seven more hot days. Satellite LST complements reanalysis diagnostics, as higher LST anomalies occur over areas under severe drought, indicating a higher control and amplification of the heatwave by surface processes and vegetation stress. These cross-cutting diagnostics increase the confidence across satellite data records and reanalysis, fostering their usage in climate applications.

## 1    Introduction

In the last couple of decades, long and extremely hot summers became more frequent over Europe (Hoy et al., 2020), with increased heat waves risk (Morlot et al., 2023). Recently, Rousi *et al.* (2022) reported that their frequency is increasing three to four times faster than in other midlatitude regions. Future climate projections have been consistently indicating the increase of both mean and extreme temperatures, with extreme heat conditions becoming more likely when more severe greenhouse gas emission scenarios are considered (Christidis et al., 2015; Lhotka et al., 2018; Amengual et al., 2014; Zhang et al., 2020; Seneviratne et al., 2021; Hundhausen et al., 2023). Although the increased radiative forcing due to increasing greenhouse gas concentrations in the atmosphere is the root cause of the general shift in global temperature distributions, their spatial patterns are not homogeneous. Over Europe, warm extremes have been linked to the presence of stationary anticyclonic systems and to atmospheric blocking events (Schaller et al., 2018; Bieli et al., 2015; Brunner et al., 2018; Garcia-Herrera et al., 2010; Chan

et al., 2022). The increasing frequency of this blocking pattern has been linked to the increasing persistence of double jet stream structures, particularly because the region between the sub-tropical and polar front jets (i.e., latitudes between 45 and 65° N) is characterized by negative wind anomalies and positive surface temperatures anomalies (Rousi et al., 2022; Kornhuber et al., 2017). Furthermore, the atmospheric circulation induced by these synoptic configurations favours the advection of warm Saharan air masses into Europe, which often compromise air quality over the affected regions due to their high dust aerosol loads (Sousa et al., 2019; Miralles et al., 2014; Díaz et al., 2017).

On top of these dynamic aspects, heatwave intensities over Europe are strongly modulated by thermodynamic effects involved in land surface-atmosphere interactions (Sousa et al., 2020; Suarez-Gutierrez et al., 2020; Miralles et al., 2019, 2014). The abovementioned typical synoptic situations favour subsidence heating, reduced cloudiness, and therefore increased net surface radiation. The excess surface energy is released as surface (longwave) radiative and turbulent fluxes, both over land and over ocean, particularly over the Mediterranean (Suarez-Gutierrez et al., 2020; Juza et al., 2022). Soil moisture availability controls the partitioning of surface turbulent fluxes, since in the case of water scarcity, all the excess energy at the surface is re-emitted mostly as longwave radiative flux and sensible heat flux, both acting to increase near-surface temperatures. Compound events of drought and extreme heat conditions are among the riskiest climate-related events for Europe, especially considering their effects on mortality, crop and forest productivity and wildfire risk (Zscheischler et al., 2018, 2020; Manning et al., 2018). The mechanisms involved in feedbacks between these processes are still under debate, not only due to the remaining uncertainties within the theoretical framework, but also due to the lack of suitable observations to support it (Miralles et al., 2014, 2019; Seneviratne et al., 2021; Barriopedro et al., 2023). This means that there is a significant spread on the heatwave metrics that are provided by different global and regional models, even at higher spatial resolutions (Petrovic et al., 2023; Lin et al., 2022; Furusho-Percot et al., 2022; Molina et al., 2020; Hundhausen et al., 2023).

According to the Copernicus Climate Change Service (C3S), the summer of 2022 over Europe was by far the warmest on record to date, with an excess of 0.4 °C with respect to 2021, the previous warmest year (https://climate.copernicus.eu/2022-saw-record-temperatures-europe-and-across-world, last access: 31 August 2023). The World Health Organization (WHO) revealed at least 15000 deaths due to the extreme heat conditions, particularly over Spain, Portugal, United Kingdom and Germany (https://www.who.int/europe/news/item/07-11-2022-statement---climate-change-is-already-killing-us--but-strong-action-now-can-prevent-more-deaths, last access: 31 August 2023). The combined effects of drought and extreme heat also led to a wide range of economic impacts, namely an overall crop loss (particularly cereal) of 9% with respect to the previous years' five-year average production (FAO, 2022), causing a generalized increase in food and grocery prices.

Near real time (NRT) monitoring of these extreme heat events as well as of the wide range of their associated variables is therefore key for timely actions from stakeholders, namely those involved in civil protection and agricultural management. Furthermore, when mega drought and heatwave events happen, there is a sense of urgency by the public and the media, particularly in what concerns their connection and attribution to climate change (Schiermeier, 2018). While such diagnostics may take weeks to perform, remote sensing provides the means for a real-time overview of the magnitude of a given event, from minutes to just a few hours after the relevant measurements were acquired. Satellite agencies, particularly the European

Organisation for the Exploitation of Meteorological Satellites (EUMETSAT), European Space Agency (ESA), Copernicus and National Aeronautics and Space Administration (NASA), are making efforts to maintain stable, accurate and long-term data records of Essential Climate Variables (ECVs; Bojinski et al., 2014), as well as to means to access digested information from those datasets. The EUMETSAT Satellite Application Facility on Land Surface Analysis (LSA-SAF; Trigo et al., 2011) provides a collection of data records related to the monitoring of land surface energy balance (including Land Surface Temperature, *LST*) and vegetation indicators from 2004 onwards. Most products are based on measurements made by the Spinning Enhanced Visible and Infrared Imager (SEVIRI) onboard the Meteosat Second Generation (MSG) series, and their production will be ensured along the next generation of geostationary meteorological satellites. Despite the relatively good temporal and spatial coverage of *LST* products over areas with significant heatwaves in the past decades (such as south-central Europe), Reiners et al. (2023) showed that these products have not been much used to study these phenomena. According to these authors, this is due to 1) the lack of long time series (i.e., longer than 30 years), 2) the absence of reliable all-sky *LST* datasets, 3) the lack of intercomparison between different *LST* timeseries and intercomparison among different climate indicators and finally, 4) the lack of validation over heterogeneous sites. However, the potential of the standard LSA-SAF clear-sky *LST* dataset for monitoring heatwaves has already been highlighted by (Gouveia et al., 2022) who derived similar heatwave diagnostics as those obtained using well established products such as MODIS LST (Wan, 2014) or ERA5/ERA5-Land reanalysis (Hersbach et al., 2020; Muñoz-Sabater et al., 2021), which have been more frequently used to study these phenomena (Zaitchik et al., 2006; Galanaki et al., 2022; Sutanto et al., 2020; Hundhausen et al., 2023; Morlot et al., 2023; Agathangelidis et al., 2022). Good et al. (2022) also demonstrated the usefulness of *LST* for climate monitoring and found good agreement between *LST* anomalies derived from the ESA Climate Change Initiative *LST* (Pérez-Planells et al., 2023) datasets and in situ 2-meter temperature anomalies derived from the ECAD/E-OBS dataset (Cornes et al., 2018). Nevertheless, the association between *LST* and 2-meter temperature anomalies under heatwave conditions is not always straightforward (Agathangelidis et al., 2022; Mildrexler et al., 2011), as they can differ substantially both in spatial and temporal extents, especially towards higher temperatures. An additional limitation on the analysis of *LST* daily timeseries based on polar orbiting satellites (Agathangelidis et al., 2022; Good et al., 2022) is that measurements over a particular area are obtained at slightly different times of day and with different viewing and illumination geometries.

The goal of this article is to illustrate the exceptionality of 2022 summer in Europe, mainly using the datasets provided by the LSA-SAF. Clouds have been identified as a caveat on the usage of this kind of dataset for monitoring heat extremes, by introducing spatial and temporal discontinuities (Reiners et al., 2023; Gouveia et al., 2022). In particular, these discontinuities hamper a correct count of the number of hot days (especially the *consecutive* hot days, which are relevant for the determination of heatwave conditions), or a correct assessment of the spatial extent of extreme heat conditions. In this work, the new all-sky *LST* product (Martins et al., 2019) is used instead. This product is based on thermal geostationary observations by SEVIRI, is available since 2004 and complements the information provided by the clear-sky *LST* product (Trigo et al., 2021), even over areas with significant cloud coverage. LSTs under cloudy conditions are estimated using a surface energy balance scheme (Barrios et al., 2024), which is used at the LSA-SAF to estimate evapotranspiration and surface turbulent fluxes along with the

cloud-sky LST. The scheme uses radiation flux and vegetation products from the LSA-SAF, H-SAF soil moisture and a few screen-level variables from ECMWF as main inputs. Thus, it overcomes the main limitations of the standard *LST* products, especially of those that have been recently used for climate monitoring (Agathangelidis et al., 2022; Mildrexler et al., 2011; Gouveia et al., 2022). By using a product that fills in those gaps using a physically-based algorithm (i.e., estimates a land

surface temperature value taking into account the changes in radiative fluxes under clouds, as well as vegetation state and soil moisture conditions), interpolations that are many times unphysical are avoided. Although clear-sky conditions are typically the norm for heatwave conditions, clouds are nonetheless frequent and ubiquitous. Comparisons of the derived information with corresponding information derived using ERA5 data are performed, not only to provide confidence to the obtained diagnostics but also to explore the physical differences between 2-meter temperature (which is the standard variable used in

extreme heat monitoring studies) and skin surface temperature (which is what is observable by satellite). Furthermore, a cross-cutting analysis of temperature, vegetation, and soil moisture anomaly patterns, all obtained using different measurement principles, is also useful for a robust assessment of their physical consistency, which, if demonstrated, improves users' trust in those datasets and further fosters their usage for climate applications.

The following section presents the data and methods followed by the results and the main conclusions of this study.

## 2    Data and Methods

### 2.1    LSA-SAF LST

*LST* corresponds to the radiative temperature of the surface "skin", i.e., the ground or the surface of the canopy over vegetated areas (Hulley and Ghent, 2019; Li et al., 2013). The LSA-SAF (Trigo et al., 2011) has been providing near real time (NRT) *LST* estimates over Europe, Africa and part of South America since 2004, based on infrared observations from the Spinning

Enhanced Visible and Infrared Imager (SEVIRI) onboard the four Meteosat Second Generation (MSG) satellites. This dataset has been extensively validated, using a set of ground stations covering a wide range of land surface conditions (Göttsche et al., 2016; Trigo et al., 2021), ensuring the compliance of the product with its requirements in terms of accuracy, uncertainty and temporal stability. However, a major limitation of that product is the fact that it is not available in cloudy situations. With this in mind, a new all-sky *LST* product was developed and is now distributed in NRT (Martins et al., 2019), the MLST-ASv2

(available in https://datalsasaf.lsasvcs.ipma.pt/PRODUCTS/MSG/MLST-ASv2/, accessed in 16 September 2023). The new product fills in the blanks in the clear-sky retrievals due to clouds, by solving a Surface Energy Balance model (Barrios et al., 2024), whose main inputs are satellite retrievals of longwave and shortwave downwelling radiative fluxes, as well as albedo and Leaf Area Index (*LAI*), all produced at the LSA SAF. Other inputs include soil moisture from the EUMETSAT SAF on Hydrology (H-SAF) and screen-level meteorological variables (including 2 m temperatures, 2 m dewpoint temperature, 10 m

winds and surface pressures) from the European Centre for Medium-Range Weather Forecasts (ECMWF). The model uses an iterative method to determine four unknows: sensible and latent heat fluxes, skin temperature and friction velocity. This skin temperature is used to fill in the gaps in the clear-sky *LST* product, while the latent heat flux is distributed as a product *per se*.

The results in the product Validation Report (Martins and Dutra, 2022) showed an overall accuracy (bias) of 0.0 K and a Root Mean Squared Difference of 2.9 K, when product timeseries are compared to measurements from 33 in situ stations over a range of land cover types and climate zones. The product is available every 30 min in NRT, with a 3 km spatial resolution at nadir (and about 4-5 km over Europe) and is then reprojected into a 0.05° regular grid. It was reprocessed from 2004 onwards using the satellite data records available at the LSA-SAF, meteorological variables from ERA5 reanalysis and a combination of soil moisture products from the H-SAF.

In this study, the daily maximum $LST$ ($LST_{Max}$) is derived from the 30-min data by taking the maximum over all timeslots from 06 to 15 UTC, when at least half the data over that period is available. This way, chances of getting an unphysical maximum are reduced. The All-Sky LST, despite being much more spatially complete than the corresponding clear-sky product, still has missing data. The product ATBD (Martins et al., 2018) mentions that the surface energy balance model used for cloudy scenes is not able to provide reliable data over inland waters, which are excluded from the analysis. There are a few situations where the model does not reach convergence after the upper limit of iterations, on which case the model also returns a missing value.

## 2.2    H SAF Soil Moisture

The Satellite Application Facility on Hydrology (H-SAF) produces, among other variables, several soil moisture datasets. However, a soil moisture data record that is fully compliant with the existing NRT products does not exist yet. Therefore, to reprocess the surface energy balance model back to 2004 and to analyse soil moisture anomalies in section 2.4, a combination of two products was used:

a)  the H141 data record (Fairbairn and de Rosnay, 2020) was used from 2004-2018 and complemented with the data record extension H142 for 2019-2020. The product is based on a land data assimilation system which assimilates scatterometer data (including ERS/SCAT and Metop ASCAT-A/B) and screen level variables (2 m temperature and relative humidity). The HTESSEL Land surface model is used to propagate soil moisture information through the soil down to the root zone.

b)  Since H141/H142 was not available from 2021 onwards, the H26 product (Fairbairn and de Rosnay, 2021) was used. This product only assimilates scatterometer data from ASCAT (A/B/C) and uses a stand-alone surface analysis derived from the 9 km operational analysis. Although H141/H142 and H26 are not identical, a comparison during an overlap period (in 2020) showed that the differences over Europe were reduced (not shown).

Soil   moisture   products   from   the   H-SAF   are   available   through   their   web   portal (https://hsaf.meteoam.it/Products/ProductsList?type=soil_moisture, accessed in 16 September 2023). In the case of the selected dataset, soil moisture is linearly rescaled between wilting point (0) and field capacity (1), defining the Soil Wetness Index (SWI). In this work, an SWI average of the first three layers (i.e., down to 1 m below the surface) from the daily data is used to compute monthly means, from which the 2004-2021 climatology and anomalies are derived.

## 2.3    LSA-SAF Vegetation Products

Vegetation plays an important role in the exchange of energy between the soil and the atmosphere (Katul et al., 2012; Van Dijke et al., 2020), by efficiently promoting the water exchange between the surface and the atmosphere. The LSA-SAF

provides several satellite derived vegetation parameters. Here, the fraction of vegetation cover (FVC, García-Haro et al., 2019) obtained from MSG observations is used to study the vegetation state as a response to the high temperatures experienced in the summer of 2022. *FVC* is the horizontal fraction of soil covered by green vegetation, ranging from 0 to 1. It is available since 2004 as a daily product (in https://datalsasaf.lsasvcs.ipma.pt/PRODUCTS/MSG/MTFVC/, accessed in 16 September 2023) and is also an indicator of drought conditions, exhibiting pronounced negative anomalies over areas under significant drought.

## 2.4    ERA5/ERA5-Land Reanalyses

To provide a synoptic context for the 2022 summer conditions (see section 2.1 below) the ECMWF ERA5 climate reanalysis (Hersbach et al., 2020) was analysed. ERA5 provides hourly data on several atmospheric, land surface and ocean parameters together with their respective uncertainties, on a 0.25˚ x 0.25˚ grid, and can be downloaded from the Copernicus Climate Change Climate Data Store (https://cds.climate.copernicus.eu/, accessed in 16 September 2023). Variables used here include 850 hPa temperature ($T_{850}$), 500 hPa geopotential height ($Z_{500}$), wind speed ($\vec{v}$) and precipitation (*precip*), for the period 1980-2022 (43 years).

ERA5-Land is a reanalysis that provides a more detailed description of the variables characterizing the continental surfaces, with a higher spatial resolution (0.1°), and is produced with atmospheric forcing from ERA5 (Muñoz-Sabater et al., 2021), relying on the ECMWF land surface model. It is therefore more accurate since surface fields such as orography or land cover are more detailed. The ERA5-Land variables used here are the 2 m temperature ($T2m$) and the skin temperature ($SKT$), which is comparable in terms of physical meaning to satellite *LST*. Although ERA5 is by far the most widely used dataset in heatwave studies, in this study $T2m$ estimates from ERA5-Land are used, as this choice allows to focus on the physical differences between $T2m$ and $SKT$, without having to consider differences introduced by comparing different modelling systems and spatial resolutions. Both $SKT$ and $T2m$ are used to derive "reference" heatwave indicators, to which the ones derived by satellite are compared.

## 2.5    Heatwave definition and Metrics

To derive monthly and seasonal anomalies, the reference "climate" is calculated by first estimating *Tx* monthly means for all months in the reference period of 2004-2021, and then taking the median of that monthly mean across the whole reference period. For JJA anomalies, the June, July and August means are averaged for each year and then the median over all years is calculated to obtain the reference "climate" for that case. When computing monthly means, it was ensured that at least 85% of the days in each month were available. This prevents spurious values in the disk edge to contaminate the monthly value.

Heatwaves are commonly characterized as a consecutive number of days when the temperature is excessively higher than normal (Sutanto et al., 2020; Xu et al., 2016; Meehl and Tebaldi, 2004). However, several authors use different definitions, which have a significant influence on the assessment of the impact of climate change on this phenomenon. In this study,

heatwaves are defined as a period of three or more consecutive days with daily maximum temperature ($Tx$) above the 90th

percentile ($P_{90}$) of the reference period 2004-2021. These days where P90 is exceeded are hereafter referred to as *hot days*. The percentile is calculated for each day of the Julian calendar considering a 31-day window around the Julian day, for all years in the reference period, based on Russo et al. (2015). The $P_{90}$ and the multi-year medians used in this study were calculated relying on a bootstrapping technique as defined by Zhang et al. (2005). This technique consists of replacing the year for which the percentile is calculated by the next year in the timeseries, except for the last year, where a mean of the previous

years' estimates is used. This kind of procedure allows avoiding the possible effects of heterogeneity between the distributions of values in the reference period and the year where the percentile is evaluated.

To facilitate comparison across different periods and regions, the magnitude of a heatwave is estimated through a standardization of the daily maximum temperature, $Tx$. The daily magnitude, $M_d$, proposed by Russo et al. (2015) is used:

$$M_d = \begin{cases} \dfrac{Tx - P_{25}}{P_{75} - P_{25}}, & \text{if } Tx > P_{25} \\ 0, & \text{if } Tx \leq P_{25} \end{cases} \tag{1}$$

Here, *Tx* may be obtained using the LSA-SAF *LST*, or either *SKT* or *T2m* from ERA5. As in Cardoso *et al.* (2019), a slight

difference to the standard index relies on the fact that the 25th and 75th percentiles ($P_{25}$ and $P_{75}$) are calculated considering all *Tx* values in the reference period, whereas in Russo et al. (2015) these were obtained from annual maxima timeseries. This means that there is one percentile value per pixel, which implies that the same temperature anomaly causes a larger daily magnitude $M_d$ over areas with less temperature variability within the reference period. In the case of $P_{50}, P_{90}$, the annual variability is still represented, to ensure that anomalies/ exceedances are evaluated against their expected values for a given

time of year.

Finally, also following Russo et al. (2015), an adapted version of the Heat Wave Magnitude Index (HWMI) is used, which is simply the sum of the daily magnitudes, $M_d$ over a given period (e.g., a single heatwave, a month or a full year), for all the heatwave days in that period:

$$HWMI = \sum_{d=d_1}^{d=d_2} M_d \tag{2}$$

where $d_1$ and $d_2$ are the Julian days between which the sum is computed. By considering the duration and intensity of heat

waves, *HWMI* allows the quantification of the magnitude of heatwaves in different periods and regions of the world (Russo et al., 2015). In this work, HWMI is always computed for the whole JJA period.

## 3    Results

### 3.1    Synoptic context

Before the analysis of the satellite data characterizing the exceptional 2022 summer over Europe, the synoptic context is
provided in this section for completeness.  ERA5 data was used for this purpose. Figure 1 shows this synoptic context for the
March-April-May period (meteorological spring, MAM, left panels) and for June-July-August (meteorological summer, JJA,
right panels). The different panels illustrate the anomalies of temperature at 850 hPa ($T_{850}$), 500 hPa geopotential height ($Z_{500}$),
as well as normalized anomalies of accumulated precipitation (i.e., where the 1981-2022 seasonal mean was subtracted from
the 2022 *precip* and then divided by the seasonal standard deviation of the whole period). As may be seen in Figure 1a and in
Figure 1c, there was an extended area of positive $Z_{500}$ anomalies over the North Sea region, covering the British Isles,
Scandinavian Peninsula and Central Europe. The $Z_{500}$ anomalies in the center of this system were above the 95[th] percentile of
the distribution of the seasonal anomalies for this period (not shown). This blocking pattern, characterized by strong subsidence
warming and relatively lower humidity, inhibited cloud formation and, consequently, induced large areas of negative anomalies
of *precip*. Moreover, this pattern was associated with an anomalous easterly/north-easterly wind regime, bringing drier
continental air into Central Europe. Transient lows from the North Atlantic were deflected from these regions by the
anticyclonic blocking. Therefore, above normal $T_{850}$ values (Figure 1a) and below normal *precip* (Figure 1c) were observed
in North and Central Europe, with areas in Southeast France showing some of the warmest and driest anomalies in the whole
reference period (highlighted as dotted areas). The JJA synoptic configuration (Figure 1b), shows that the atmospheric blocking
pattern over central Europe persisted and even aggravated across the summer, with the centroid of the anomaly located more
towards Central Europe (when compared to the MAM configuration). Once again, it is characterized by exceptionally high
positive $Z_{500}$ anomalies and associated with a prominent anomalous easterly wind towards central/western European countries,
such as France, Germany and Italy. Persistent warm and dry advection continental air masses from Eastern Europe contributed
to the exceptionally high $T_{850}$ anomalies over France, Italy and parts of Spain and Germany. Lack of *precip* was also observed
over the Iberian Peninsula, Germany and the British Isles (Figure 1d), with many areas where it was below the 10[th] percentile
(shown as dotted areas). This configuration of higher-than-normal temperatures in spring and summer and overall lack of
rainfall, especially during springtime, lead to the intensification of the widespread drought event that started in early spring
and lasted throughout the entire summer.

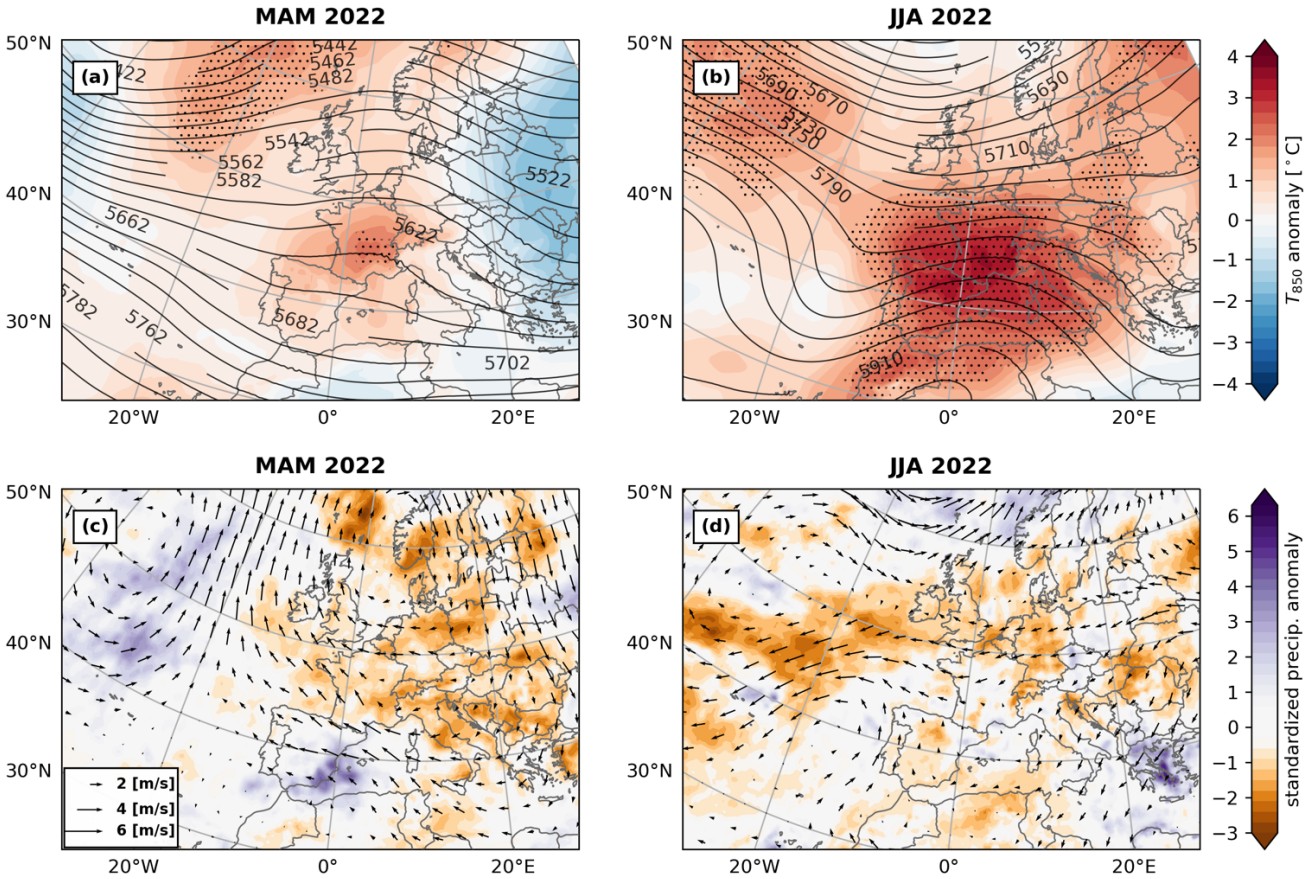

**Figure 1 - Panels representing the anomalies of the synoptic atmospheric configuration over Europe for two seasons in 2022, as given by ERA5: MAM (panels (a) and (c)) and JJA (panels (b) and (d)). Panels (a) and (c) show $T_{850}$ anomalies (in colour), and $Z_{500}$ (black contours). Dotted areas denote areas where $T_{850}$ was above its 95th percentile. Panels (b) and (d) show the normalized anomaly of accumulated precipitation (in colour) and $\vec{v}$ anomalies (black arrows). Dotted areas denote areas where *precip* anomaly was below the 10th percentile. All anomalies were computed with respect to the 1981-2022 reference period. Arrows are spaced 2°x2° for the sake of readability.**

## 3.2    LST anomalies and Comparison with ERA5

We start the analysis by focusing on the summer LST climatology and anomalies for the years 2018, 2019, 2021 and 2022 (Figure 2). These were among the recent years with largest JJA anomalies over Europe. The 2004-2021 climatology is represented in Figure 2a and shows that higher $LST_{Max}$ are observed around the Mediterranean, except for coastal areas that are more prone to the occurrence of low clouds (such as west Iberia). Panels (b), (c), (d) and (e) show the JJA anomalies for 2018, 2019, 2021 and 2022, respectively. The 2022 $LST_{Max}$ JJA anomaly was much stronger when compared to other years, both in terms of the anomaly magnitude and of its spatial extent. Seasonal anomalies of 3-5 °C were observed over most of Central Europe, in an area extending from Northern Spain to the British Isles and to Eastern Germany. Large areas of France exhibited seasonal anomalies up to 7-8 °C. The area with the largest anomalies was over Hungary, where LSTs were 9.5 °C

above normal. Over the considered domain, 2022 showed an area-averaged JJA anomaly of 2.2 °C (where the anomaly was

weighted by the area of each pixel), while the remaining years with the highest area-averaged JJA anomalies in the data record, show values of 1.1°C (2018 and 2019) and 0.8°C (2012).

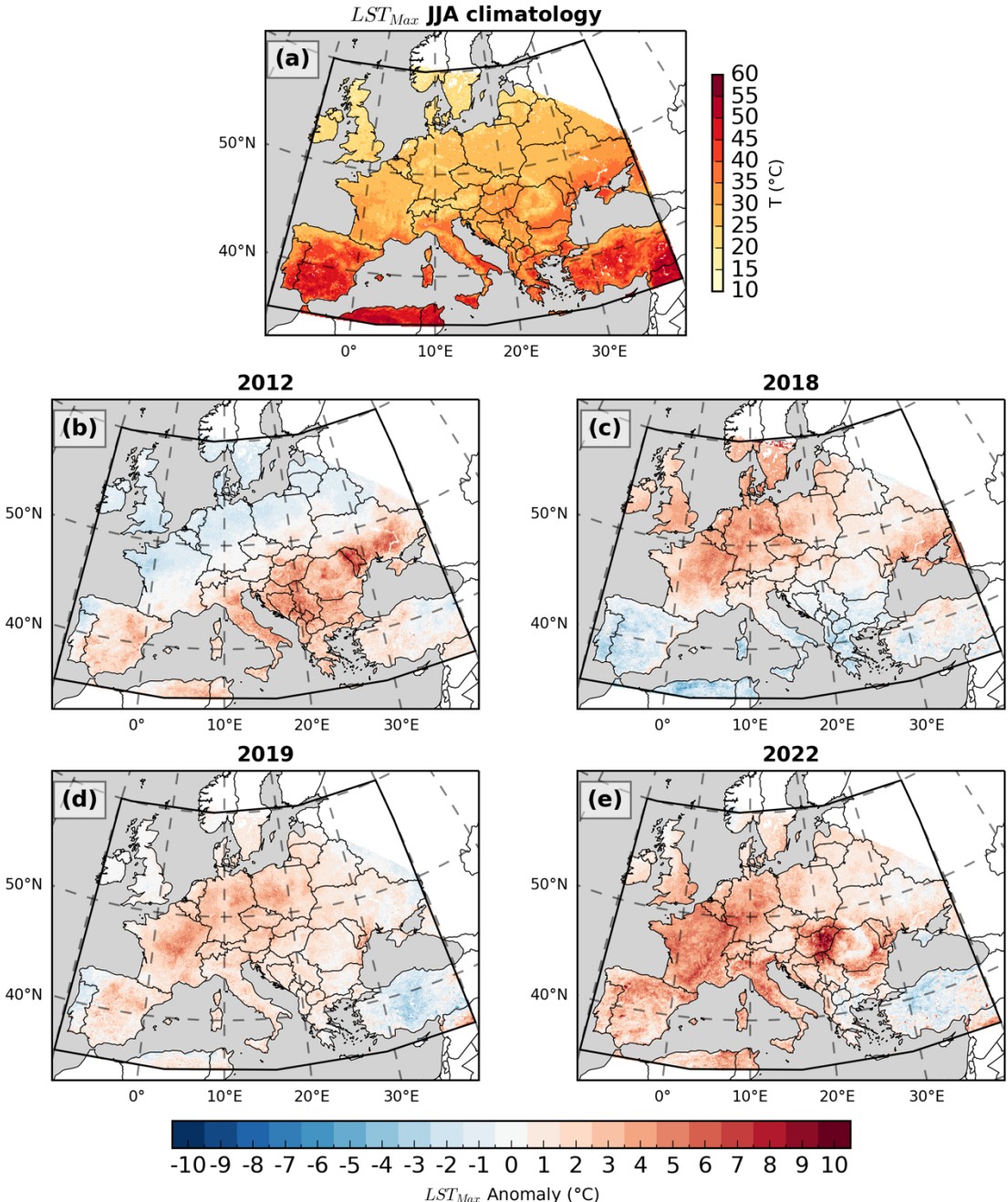

**Figure 2** - (a) JJA median of $LST_{Max}$ for the period 2004-2021, while (b, c, d, e) are seasonal $LST_{Max}$ anomalies for 2012, 2018, 2019 and 2022.


Figure 3 shows the evolution of monthly anomalies of the three summer months (JJA). In June, a general pattern of positive temperature anomalies is evident throughout Europe, with values ranging around 3-4 ˚C, and in Hungary, northern Spain and Italy there were anomalies of up to 8 ˚C. Both Northwest Iberia, the British Isles and countries bordering the Eagean Sea exhibited slightly colder-than-normal temperatures. In July, the anomalies over Hungary and Romania suffered a very strong increase, with values around 7-10 ˚C, while in Central/Western Europe there were anomalies ranging between 3-6 ˚C. In North-Eastern Europe and Turkey, the anomalies were negative, with temperatures about 1-3 ˚C lower than normal. In August, the anomaly was the most intense of the three summer months, with an average value of around 3 ˚C above the climatological reference. The most affected areas by anomalously warm conditions were Central Europe and Hungary, with the monthly anomaly for these areas ranging from 8-10 ˚C. The pattern of positive anomalies extended into Eastern Europe, where the monthly anomaly ranged around 2-5 °C. South Balkan countries and Ukraine were left out of the general pattern of very high temperature anomalies in August.

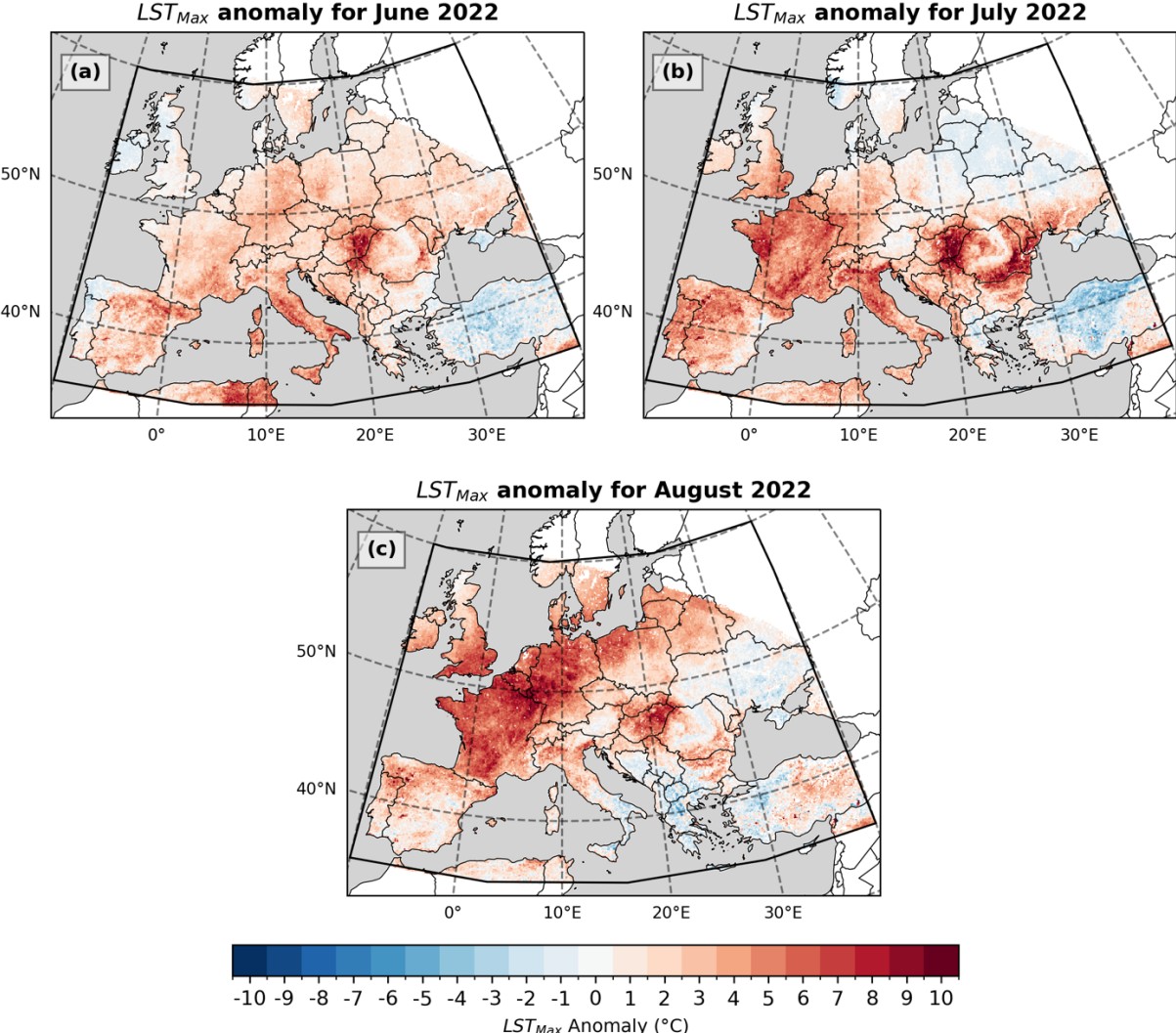

**Figure 3 - $LST_{Max}$ monthly anomalies for (a) June, (b) July and (c) August 2022 over Europe.**

Although there are a few recent climate assessments made using remotely sensed LST (Wang et al., 2022; Good et al., 2022; Gouveia et al., 2022), this kind of data is not typically used to derive this kind of climate monitoring information, and therefore a comparison with more standard datasets is relevant in the context of this work. In Figure 4, a comparison between the anomalies shown in Figure 3 and the corresponding anomalies using ERA-5 Land $SKT$ and $T2m$ is shown. All anomalies used the same reference period and calculation methodology. While $LST$ and $SKT$ are highly comparable in terms of their physical meaning, $T2m$ results mostly from assimilated surface meteorological observations, while $SKT$ has a stronger model weight. $LST$ for clear-sky situations (which typically prevail in heatwave conditions) is derived from thermal infrared brightness temperatures. For cloudy skies, a surface energy balance model is employed, which is mostly based in optical and infrared satellite information, but also relies on screen-level data from ERA5 for the estimations of surface fluxes. Similarly, ERA5-

Land *SKT* is also derived from a surface energy balance driven by ERA5 fluxes and screen-level data, modulated by the land surface characteristics (e.g., vegetation cover) and conditions (available soil moisture). Therefore, it is influenced by ERA5 errors, such as errors in cloud fraction, or errors in the representation of the physiographic fields such as vegetation cover/LAI (which are static) or sub-surface conditions (soil moisture).

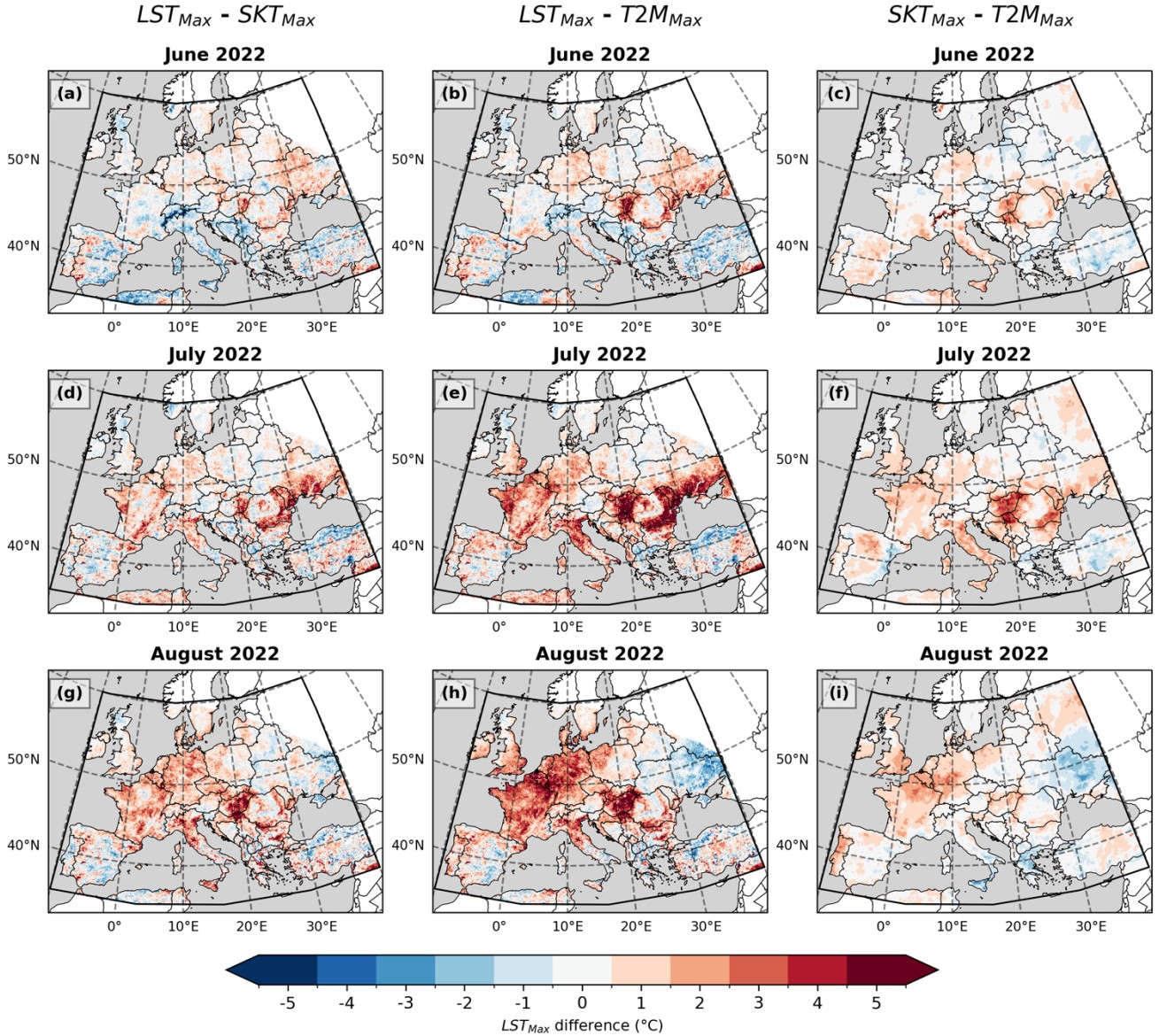

**Figure 4 – Comparison between $LST_{Max}$ monthly anomalies and the corresponding anomalies using reanalysis *SKT* (left) and $T2m$ data (right). Comparisons are made for June (a, b), July (c, d) and August (e, f).**

For instance, temperature anomalies over burned areas are generally higher when they are determined based on *LST* observations, than when they are based in *SKT*, since the relevant information in the physiographic fields is not included in

ERA5-Land (e.g., surface emissivity, albedo, vegetation cover, etc.). Some of these fire scars are visible in Figure 3 (e.g., over the Iberian Peninsula). Close inspection of pixels roughly corresponding to burned areas associated to fires occurred in July 2022, namely in northwest Spain (Castilla and Leon) and in the south (Andalusia), reveals LST-SKT mean differences of up to 14°C in the August maps. $LST$-based anomalies are generally comparable to $SKT$-based anomalies in June. A more pronounced negative difference is observable over the Alps, and more pronounced (3-5 °C or more) positive differences were observed over Western inland Iberia, Hungary, Romania, and Ukraine. Throughout the summer, these positive difference patterns intensified, and positive differences rose all over Central and Mediterranean Europe, with August being the month where the overall differences were higher, reaching 3 to 5 °C or more.

Regarding the differences between the $LST$-based and the $T2m$-based anomalies in June, the spatial patterns are similar to those of the $LST$ difference with $SKT$ (panels a, b of Figure 4). However, there some difference worth noting: *(i)* absence of a pronounced negative difference over the Alps, *(ii)* a more consistent positive difference over Central Europe and *(iii)* higher differences over the regions where the pronounced positive differences were identified in the $SKT$ map. For July and August, the differences become much higher, exceeding 5 °C over Hungary, Central Europe, Northern Italy and areas around Northern Black Sea. However, in August, a pattern of negative differences up to -3 °C raised in the easternmost parts of the domain (panel f of Figure 4).

Regarding $SKT_{Max} - T2m_{Max}$, one should note that these are entirely produced by reanalysis alone. The comparison reveals that thermal contrasts between $SKT_{Max}$ and $T2m_{Max}$ are much smoother than those between $LST_{Max}$ and $T2m_{Max}$. Since the surface sensible heat flux is proportional to this difference, this suggests that sensible heat fluxes are weaker in ERA5-Land under extreme heat conditions than compared to observations (i.e., $LST_{Max} - T2m_{Max}$ differences), although other model parameters might play a role in the sensible flux modulation (e.g. surface roughness).

In Figure 5, the temperature differences are further analyzed as a function of the absolute $LST_{Max}$. Their behaviour is consistent across the absolute $LST_{Max}$ range. For instance, for lower $LST_{Max}$, both differences are small and negative. A large part of this can be explained by persistent clouds, which if undetected, could introduce a negative bias in LST (Martins and Dutra, 2022; Trigo et al., 2021; Martins et al., 2019). These situations are however relatively infrequent. For mid-range $LST_{Max}$, differences are generally positive, with larger $LST_{Max} - T2m_{Max}$, especially in July when they reach around 2°C. For $LST_{Max}$ around 45-55 °C, temperature differences are relatively lower, but they increase again for very high $LST_{Max}$.

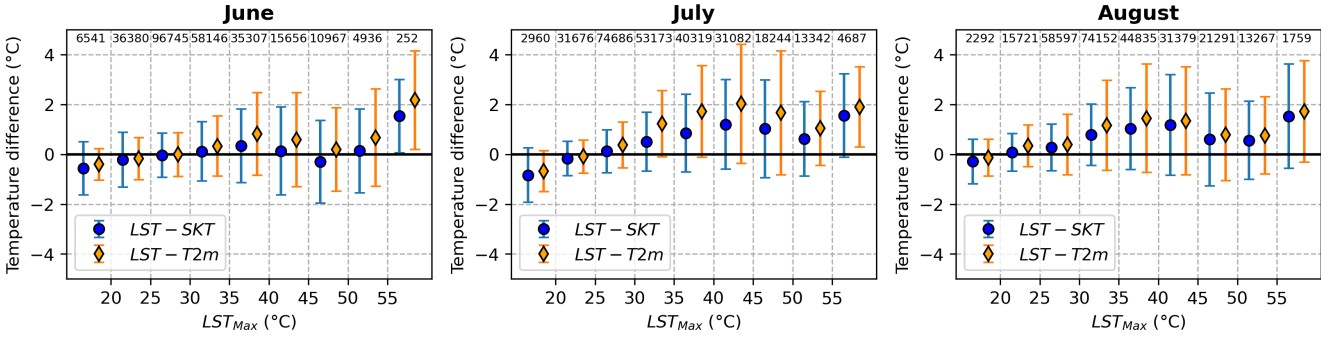

**Figure 5** – Mean differences between $LST_{Max}$ and $SKT_{Max}$ (orange, diamonds) anomalies and between $LST_{Max}$ and $T2m_{Max}$ (blue, circles) anomalies as a function of mean $LST_{Max}$, for June (left panel), July (center panel) and August (right panel). On top, the number of pixels used in the calculation. Whiskers represent the standard deviation over each interval.

Therefore, despite the good correlations between *LST* and *T2m* found by Good et al. (Good et al., 2022), these results show that there is a wide range of situations where these temperatures may be very different.

### 3.3    Number of Hot Days and HWMI

Following the previous results focusing on the seasonal and monthly anomalies, we now assess several aspects of heatwave related metrics. Figure 6a presents the number of Hot Days in JJA, using $LST_{Max}$. The most striking result is that only a few regions have less than 10 Hot Days, which include East Iberia and the Southern Balkans. In contrast, large areas over the whole West-Central Europe area had more than 40 Hot Days (and even more than 50). The Heatwave Magnitude Index (HWMI) for JJA shown in Figure 6b provides a cumulative view of the extreme heat conditions for each pixel. The differences between Figures 6a and b stem from the requirement of at least three consecutive hot days for HWMI to be positive. Heatwave conditions were particularly severe (i.e., with HWMI values up to 120) in Northeast Portugal and Spain, Southeast France, Hungary and Slovakia, parts of Romania and in a lesser extent (i.e., with HWMI values between 60 and 100) Northwest France and Luxembourg. Regions such as Southeast Spain, Scotland, Austria, Czech Republic and the southern Balkans were not severely affected by damaging heat conditions in the summer 2022.

The impact of using *SKT* or *T2m* instead of $LST_{Max}$ to derive the heatwave diagnostics is assessed in panels c-f of Figure 6 showing the differences of the indices. In general, the patterns are similar to those observed in Figure 4, translating the fact that the physical differences between these three variables necessarily impact these heatwave diagnostics. In central Iberia, $LST_{Max}$ reveals a pattern with up to less 40 hot days when compared to *SKT* and *T2m*, consistent with negative difference in thermal anomalies with respect to the ERA5 variables (see Figure 4). In Central Europe, by using $LST_{Max}$, up to 20 more hot days were detected and increases around 20-40 in HWMI were observed. The largest differences are over Northern Italy, Hungary and East Romania, where there are up to more 40 Hot Days and differences of up to 60 in HWMI, with respect to both the ERA5-Land variables.

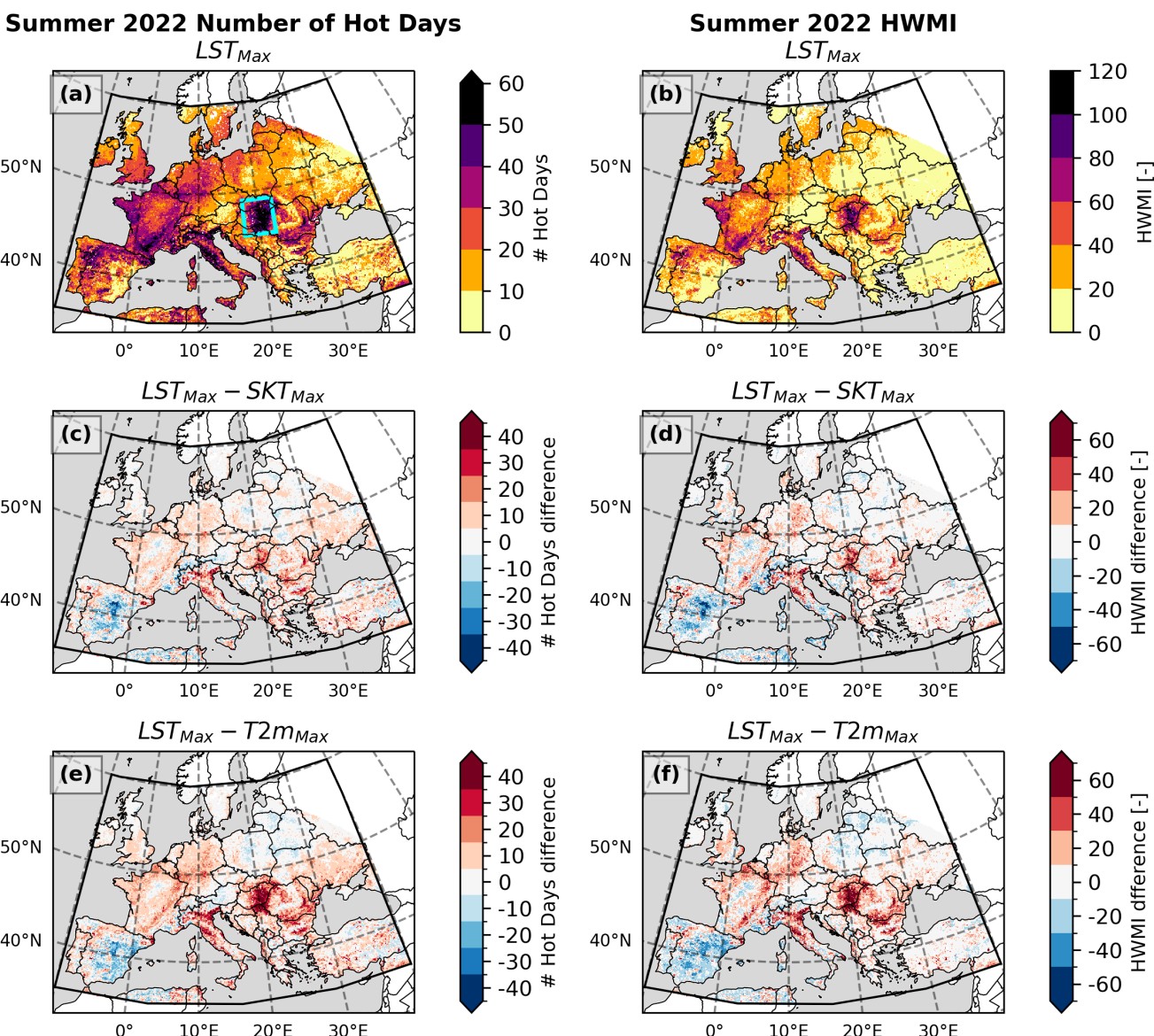

**Figure 6 – (a) Number of JJA hot days detected using the $LST_{Max}$ (i.e., days when $LST_{Max} > P_{90}$). (b) Total JJA HMWI derived with $LST_{Max}$. (c, e) Differences between the number of Hot Days obtained with $LST_{Max}$ and with $SKT$ and $T2m$, respectively. (d, f) Difference to the $SKT$-based HWMI and T2m-based HWMI, respectively. The blue square in (a) denotes the area used for the extraction of timeseries data which are analysed below.**

In Figure 7, a timeseries view of the mean behaviour of the heatwave diagnostics within the box in Figure 6a is shown to illustrate some methodological aspects. Large day-to-day $LST_{Max}$ variability is observed over this region, with temperature drops of more than 10 °C over a couple of days (e.g., from 29th to 31st July), mostly caused by cloudy conditions and advection of colder airmasses into the region. Even so, 40 hot days were observed, 33 of which belonged to heatwave periods (red

circles). These counts are of course very sensitive to the $P_{90}$, shown as a dashed line in the top panel in Figure 7. Both the

number of days in rolling window and the number of years used to determine the percentile influence the heatwave diagnostics, because on one hand a larger moving window tends to lower the summer $P_{90}$, and on the other hand, including more earlier years (which are colder due to climate change) would also lead to a decrease of these $P_{90}$ values. Therefore, if any of these options would have been considered, the observable heatwaves in the end of June, beginning and mid-August could last for slightly longer and the two last hot days observed in August could have been part of a heatwave, and the overall HWMI would

be slightly higher for the region.

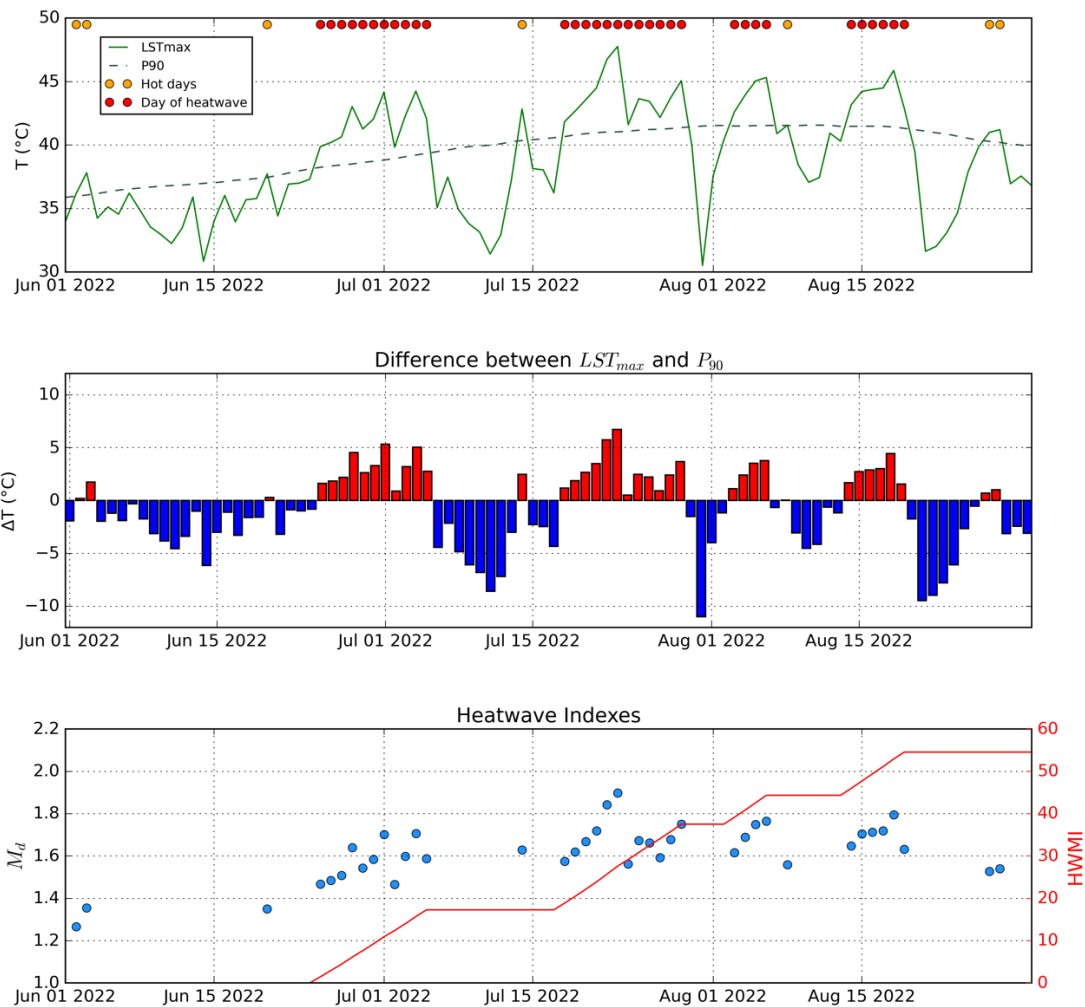

**Figure 7** – **(top)** Evolution of $LST_{Max}$ (green curve) and the respective $P_{90}$ (dashed curve). Hot days are marked as a yellow circle at the top; if they belong to a heatwave (set of 3 or more consecutive days), they are marked as a red circle. **(middle)** Explicit differences between $LST_{Max}$ and the $P_{90}$. **(bottom)** Daily heatwave magnitude, $M_d$ is in blue and the accumulated HWMI is in red, with values

in the right axis. All data are area averages from the blue box in Figure 6.

### 3.4 Vegetation Anomalies / Soil moisture anomalies

In this section, independent remote sensing datasets are explored for the study period with a twofold motivation: *i)* to assess the extreme heat conditions in the context of the drought conditions (see Figure 1) and *ii)* to identify potential causes for the differences between the $LST_{Max}$ and ERA5-Land diagnostics observed in the in the previous sections. $FVC$ measurements are obtained through optical imagery, made by SEVIRI on MSG. This is the same instrument used to produce the clear-sky $LST$s (also providing the main inputs for its cloudy scenes), although $LST$ relies on infrared information rather than visible and near-infrared like in the case of $FVC$. $FVC$ quantifies the fraction of each pixel that is occupied by green vegetation and responds to soil moisture and surface net radiation anomalies with a delay related to plant physiology. $SWI$ is obtained through scatterometry data (i.e., radar) obtained by polar orbiting instruments (such as ASCAT on Metop) and consists of an index quantifying how close root-zone soil moisture is to soil field capacity ($SWI = 1$) or to plant wilting point ($SWI = 0$). $SWI$ is used as input to the surface energy balance model that is used to derive cloudy sky $LST$. However, it can be inferred that most of the retrievals under heatwave periods are made for clear sky. Therefore, it can be assumed that $LST$, $FVC$ and $SWI$ are mostly independent from each other.

In Figure 8 the monthly $FVC$ anomalies and the monthly anomalies of the $SWI$ index for June, July and August 2022 are shown. In June, most of Central Europe shows small positive anomalies of $FVC$. Major exceptions with strong negative anomalies are France and Northwest Iberia, eastern Hungary, and Italy (with smaller values). The $SWI$ anomaly patterns are not necessarily similar, but broadly correspond to the anomalies in accumulated precipitation (cf. Figure 1), with some exceptions. Over Switzerland, the MAM precipitation anomaly was among the highest over the reference period, while the $SWI$ anomaly is negligible. Over Germany and Poland, negative $SWI$ anomalies are observed, which are associated to the very low precipitation over the region, but their expression in $FVC$ only becomes evident from July onwards. Eastern countries exhibit strong positive $FVC$ anomalies and positive $SWI$ anomalies.

The arc-like feature covering Hungary, Serbia, Romania, and Moldova is consistent among precipitation, $LST_{Max}$, $SWI$ and $FVC$. This is also the region where differences between $LST_{Max}$-based heatwave diagnostics and those derived from $SKT_{Max}$ (and in a lesser extent, $T2m_{Max}$) are larger. The positive vegetation anomaly in the eastern parts of the domain is also consistent with the negative $LST_{Max}$-$T2m_{Max}$ differences over that area.

This consistency among different remote sensing (and reanalysed) products, obtained via different instruments and measurement principles, suggests problems in the representation of these variables in ERA5-Land. In fact, since vegetation dynamics is prescribed as static information in ERA5-Land, the reliability of some surface variables (such as surface temperature) can be questioned when these strong vegetation anomalies are in place (Johannsen et al., 2019; Nogueira et al., 2020, 2021; Duveiller et al., 2022). If dynamic vegetation was prescribed in ERA5-Land, a negative anomaly in $FVC$ would imply *a)* a decrease in latent heat release since the surface cannot evaporate water so efficiently and *b)* a reduced sensible heat flux, since roughness is reduced when vegetation cover decreases. Both these effects would act to increase $SKT$ (especially the surface roughness increase), considering that in that situation soil moisture is assimilated (thus already implying reduced

evapotranspiration). Under these circumstances, the only effective way to compensate the excess surface net radiation is to

410 increase thermal longwave emission (i.e., increasing skin temperature).

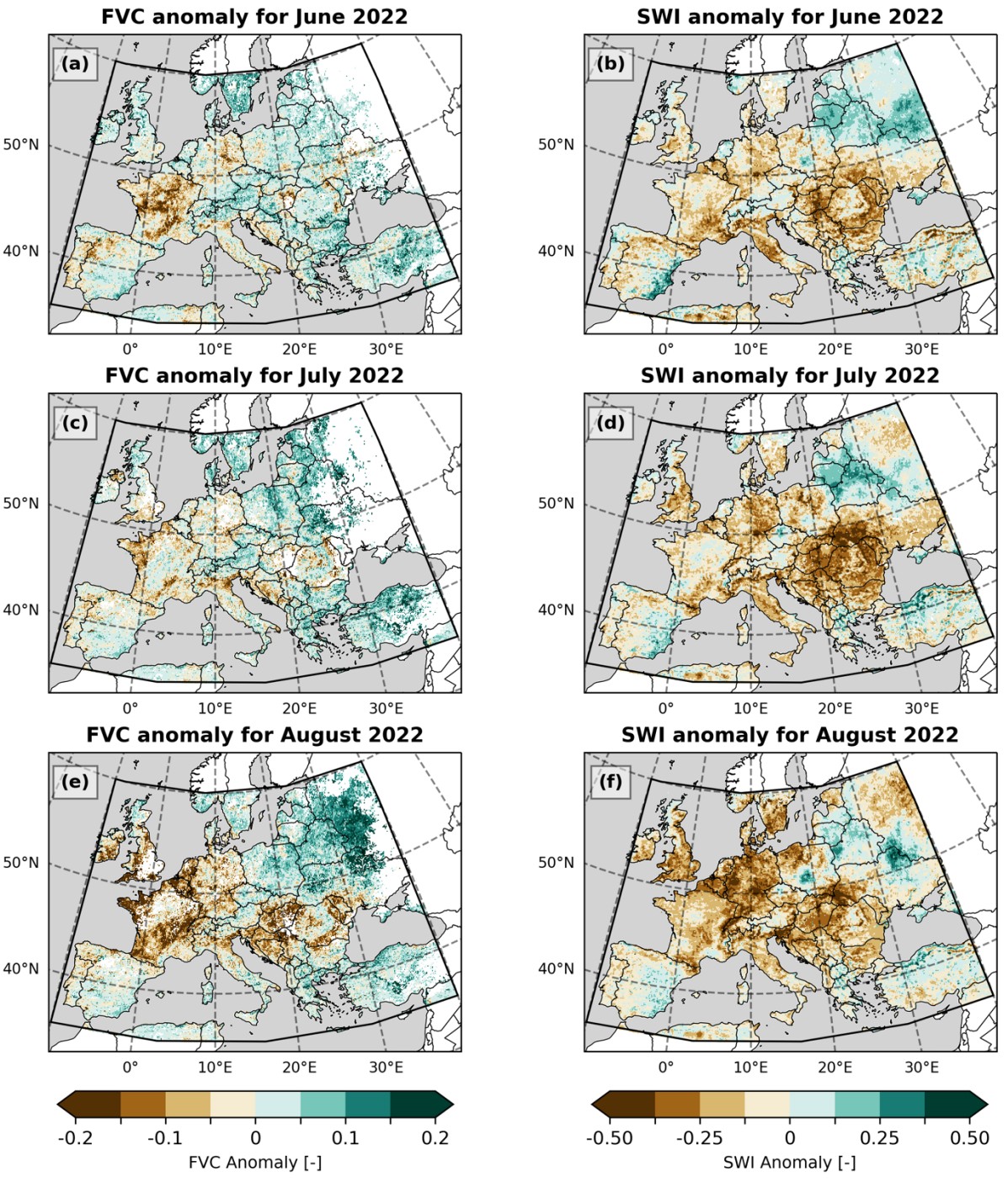

**Figure 8 – (left)** *FVC* **monthly anomalies and (right) SWI, for June (top), July (middle) and August (bottom). Reference period is 2004-2021.**

Thermal anomalies observed from satellite and those obtained from ERA5-Land *SKT* would then be more similar (i.e.,
differences in Figures 4 and 5 would be much lower). These results not only highlight the added value of using *LST* for
heatwave monitoring instead of more standard datasets such as ERA5 but may also contribute to identify ways of improving
ERA5, especially its surface scheme.

### 3.5 Exceptionality of the 2022 heatwave

To finalize the results, we show evidence of the exceptionality of the 2022 heatwave magnitude and spatial extent. Figure 9
shows a time series of the proportion of European land area affected by daily magnitudes greater than two, from June 1st to
September 1st, together with the corresponding data from the individual years from 2004 to 2023. For 22% of the summer days
in 2022, the proportion of land area occupied by $M_d > 2$ was the largest among all years. These days occured mainly in the
middle of July and for seven days in a row in mid-August. Other years like 2018, 2019 and 2015 also had large periods where
a significant percentage of European land area was under extreme heat stress.

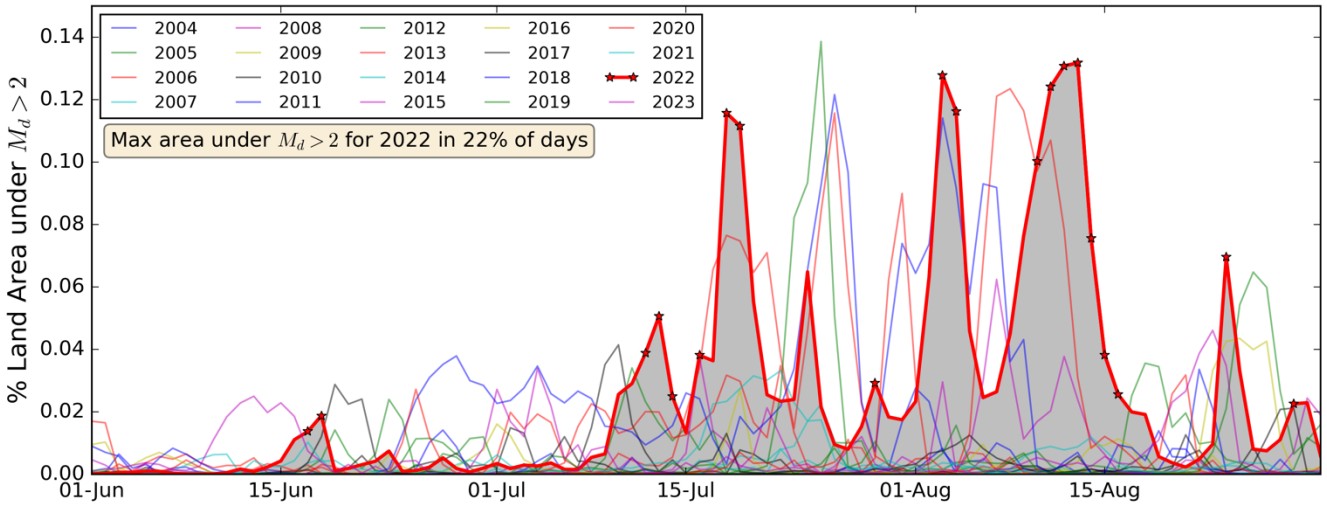

**Figure 9 – Time series of the percentage of land area affected by $M_d > 2$, from June 1st to August 31st. The red bolder curve represents 2022 data, while other colors represent the same variable for the other years in the data record. Stars mark the days where the area where $M_d > 2$ in 2022 was the greatest over all years.**

In Figure 10, all the summers in the LST data record (2004-2023) are ranked in terms of four different heatwave metrics: 1)
spatial average of the seasonal LST anomaly, 2) average area under extreme heat conditions (i.e., $M_d > 2$), the mean number
of hot days, and 4) spatial average of the summer HWMI. There are considerable disparities between the four rankings,
reflecting the way different characteristics under analysis impacted Europe in each year. Nevertheless, our results show that
2022 was remarkably exceptional, independently of the ranking criterion, followed by 2018. As already discussed in section
2.2, in terms of mean $LST_{Max}$ anomalies, 2022 ranks in first with a mean anomaly of 2.2 °C, followed by 2019 and 2018 with
a mean anomaly of 1.1 °C. The coldest year was 2004, with a mean anomaly of -1.2 °C. Even in the context of a general
increase of these mean anomalies over Europe, 2022 stands out as truly exceptional. Regarding the mean area occupied by

extreme heat conditions (i.e., the time average of the curves in Figure 9), a very strong increase has been observed since 2018, with the top three years being 2022, 2018 and 2020. In 2022, more than 2% of Europe was under extreme heat conditions, on average, and a similar picture happened in 2018. In 2020 that value was just over 1%, although with an overall lower mean temperature anomaly (ranking in 6[th] in the left panel). The year with the smaller average area covered by extreme heat conditions was 2008, with only 0.09%. As for the mean number of Hot Days, again 2022 stands out in the 1[st] place with nearly 21 hot days, on the JJA average for every land pixel over Europe. This is a very large difference towards the 2[nd] in the ranking, since in 2012 around 14 hot days on average were observed. An average of 3.7 hot days were observed in 2004, the last on the ranking. Finally in terms of area averaged HWMI, a mix between the information in the previous rankings is observed, but with an evident similarity to the number of hot days ranking. This reinforces the idea of HMWI being an index combining all the relevant information on heatwaves, namely their magnitude and their temporal and spatial extent.

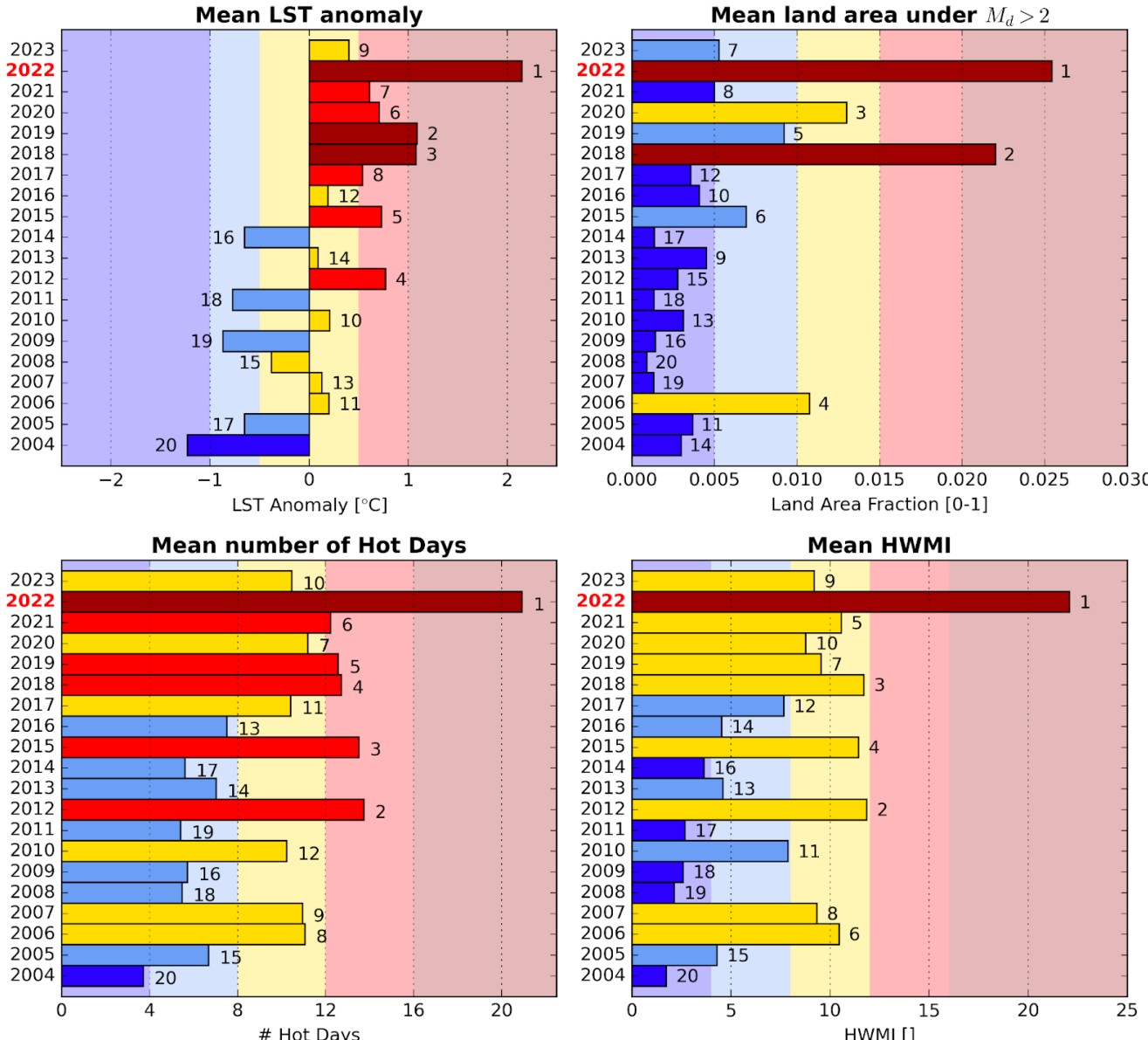

**Figure 10 – Ranking of summers over the study period according to (left) their mean $LST_{Max}$ anomaly, (middle) the average fraction of area covered by extreme heat conditions ($M_d > 2$) and (right) area-averaged HWMI. Colours are mainly for illustrative purpose, where each year was classified according to the severity associated to each parameter (from less severe in blue to extremely severe in dark red).**

Another way of inspecting the exceptionality of the 2022 extreme heat conditions is to look for areas where new temperature records were set, which is illustrated in Figure 11. In the JJA anomaly, new maxima were set for large areas of Northern Iberia, France, Southern Germany, Italy and Hungary (in bright cyan). These areas have strong signals over the individual monthly maps as well. The 2018 (dark grey) heatwave still holds the record for large parts of North Central Europe (Hoy et al., 2020),

while the 2010 (red) heatwave set the overall record over Russia (Barriopedro et al., 2011). In the June map, the 2019 (light grey) heatwave introduced records for this month over a large part of North Central Europe, while 2023 (light blue) set new records for Northern France and the Benelux area. In July, 2022 (bright cyan) set new monthly records over South Central Europe, 2006 (orange) set the July record for large areas from France to the Baltic countries, while the in the north Central

Europe the record was set by the 2018 (dark grey) heatwave. In August the year 2022 set new records for large areas from northwest Portugal, to France and the British Isles up to the Baltic countries. The 2015 (light brown) heatwave still holds the record for August over areas such as Poland, Belarus and west Ukraine. It is worth noting that 2003 set JJA temperature records over large areas of Western Europe (Sousa et al., 2020; Lhotka et al., 2018), but since that year is not within the period covered by the LSA-SAF LST dataset, it does not show up in Figure 11. This can be regarded as a caveat of this dataset, since due to

its relatively short length, it does not allow a full picture of the most relevant extreme heat events over Europe for the past couple of decades.

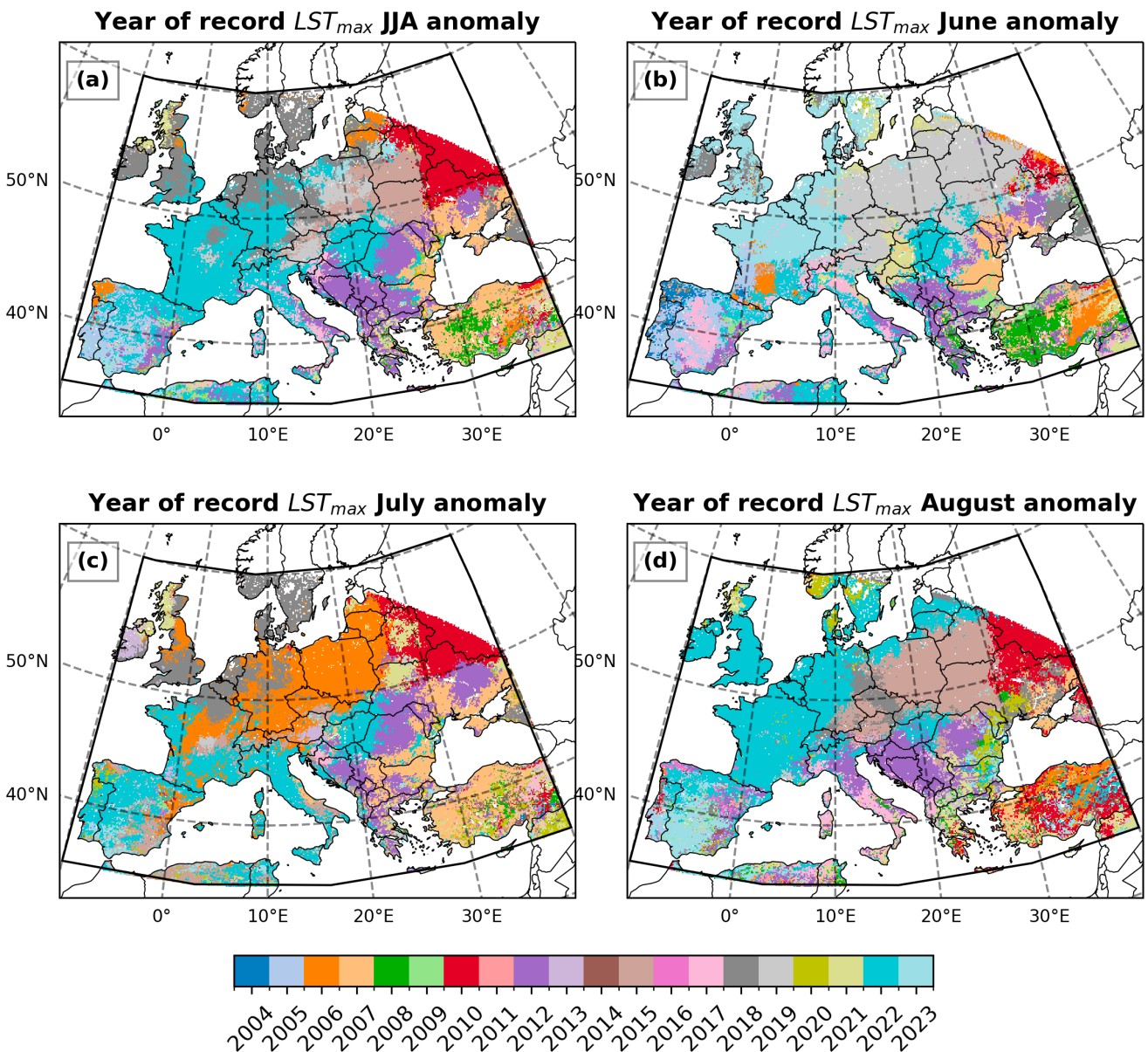

**Figure 11 – Year where the record maximum average $LST_{Max}$ occurred for the periods (a) JJA (b) June (c) July and (d) August.**

## 4    Conclusions

The year 2022 has been exceptional for Europe in terms of spatial and temporal extent and in terms of the magnitude of the heat extremes that affected the continent. The dominant synoptic configuration caused unprecedented blocking, subsidence heating and warm advection into the continent, as well as a prolonged record drought from Spring onwards. Although some

uncertainties persist, a theoretical framework is already able to broadly explain the broad mechanisms involved in the role of the surface-atmospheric coupling in the modulation of these events. The combination of these ingredients has been key in

setting the magnitude of the 2022 as well as for other recent mega-heatwaves.

The analysis presented in this study showed that the summer extreme heat conditions was unprecedent, considering the relatively short reference period. In particular, the analysis showed that this year exhibited the largest summer temperature anomaly (about 2.2°C above the reference period median), the largest average area under extreme heat conditions (just above 2.5% of Europe was on average under extreme heat conditions – $M_d > 2$) and the largest mean number of hot days (average

of 21 days). The combined effect of these factors is accounted for by the HWMI, which also translates the exceptionality of the 2022 summer over Europe, with an average value of 22 which was followed in the ranking by 2012 and 2018 with summer HWMI of 11.9 and 11.7, respectively. This study also showed that despite some areas in June and August had their highest monthly anomalies in 2023, the 2022 summer as a whole remains as exceptionally warm, ranking in first in all considered metrics.

This study also highlights the importance of looking into $LST$ as a complement to the information provided by $T2m$. The main source of $T2m$ are surface meteorological stations (point data) which may then be interpolated into gridded datasets (such as E-OBS) or assimilated into and Earth System Model or Reanalysis (such as ERA5). These methods introduce interpolation and model errors into the spatially continuous $T2m$ fields. For satellite observations, spatial coverage is much higher, and therefore spatial interpolation errors are mostly absent (although there are relatively small uncertainties associated to

geolocation and regridding from the original satellite observations to regular grids such as those used in this study). Furthermore, $LST$ is more directly linked to the surface energy balance and provides extra information when compared to $T2m$. In this work, it was shown that when there are strong drought conditions linked with vegetation anomalies (i.e., low $SWI$ and low $FVC$), differences between $LST_{Max}$ and $T2m$ anomalies are amplified, and therefore these results highlight that these observed differences may be used as proxies for surface-atmosphere coupling metrics. It should be noted however that

$SWI$ is used as input to the cloudy sky $LST$ retrieval. However, most of the retrievals under heatwave conditions are made under clear skies, so most of the $LST$ signal is coming from the infrared retrieval, and not from the surface energy balance scheme used for cloudy sky retrievals.

Given the physical similarities between them, the comparisons of $LST_{Max}$ and $SKT_{Max}$-based metrics further reinforced the confidence on the former. Not only because they compare relatively well in general, but because when they do not, there are

plausible reasons for it. ERA5-Land does not rely on dynamic vegetation information and therefore strong negative anomalies such as those reported here over Central Europe and over the Hungary-Romania region are not well represented by the model. This means that surface roughness and evapotranspiration efficiency are too high in the model, leading to colder $SKT$s when compared to the observed $LST$s. On the other hand, the cross-cutting analysis comparing $LST$, $FVC$ and $SWI$ anomalies shows a remarkable physical consistency between the observed patterns, thus reinforcing confidence in these datasets. This kind of

analysis is key to foster their usage for all sorts of climate applications, thus increasing user uptake of these datasets.

New perspectives on climate monitoring are allowed by the increasing availability of high-quality satellite data records. Although the MSG-based *LST* data record is not still at the stage of being used to derive fully compliant climate normals (which typically use 30 years of data), it already provides useful perspectives to complement the study and monitor of decadal surface temperature variability. These include the study of surface-atmosphere coupling within extreme heat events based in
observations and diagnosing caveats in more standard datasets used for the monitoring those events.

**Author contributions**

Conceptualization and Methodology by JM, RMC and SC. Data curation by ED and JM. Formal analysis by JM, SC, CP, RMC and ED. Original draft preparation by JM, RMC and SC; review and editing by all authors.

**Data Availability**

All the datasets used in this study are openly available. The MLST-ASv2 and FVC products can be downloaded from the EUMETSAT LSA-SAF Data Service (https://datalsasaf.lsasvcs.ipma.pt/), and the H-SAF soil moisture was obtained through their web portal (https://hsaf.meteoam.it/Products/ProductsList?type=soil_moisture) in The ERA5 data was retrieved from the Copernicus Climate Change Climate Data Store (https://cds.climate.copernicus.eu/). All datasets require a simple registration
before they can be downloaded.

**Acknowlegements**

This work was performed within the framework of the LSA SAF (http://lsa-saf-eumetsat.int) project, funded by EUMETSAT. Data from ERA5 and ERA5-Land were generated using Copernicus Climate Change Service information [2022]. C. P. acknowledges the support from project FRESAN - FED/2017/389-710, financed by the European Union and managed by
Camões I.P.. R.M.C. would like to acknowledge the financial support from Fundação para a Ciência e a Tecnologia, I.P./MCTES through national funds (PIDDAC) – UIDB/50019/2020 – Instituto Dom Luiz and the project LEADING— PTDC/CTA-MET/28914/2017, funded by Fundação para a Ciência e a Tecnologia, I.P./MCTES through national funds (PIDDAC).

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
