# Peer review of "A satellite view of the exceptionally warm summer of 2022 over Europe"

_EGUsphere, 2023_

## Author Comment (AC1)

RC2
**Overview**

The manuscript from Martins and colleagues makes an exhaustive description of the 2022 heatwave in Europe with a specific focus on how land surface temperatures (LST) from geostationary satellite can provide valuable information to describe such event. The manuscript is well-written and well-documented. It may sometimes reads a bit too much like a weather report rather than a scientific paper, and it does not bring major novel scientific insights, but overall is informative enough to merit publication if some points are considered.

Dear Gregory Duveiller, thanks a lot for the generally positive feedback. We appreciate the time you took to review our paper and your valuable comments.

The big question is actually 2023. It is a pity that the process of scientific analysis, writing and reviewing takes so long, and I fully understand that the authors had the intention to focus on 2022 before the summer conditions of 2023 came about. But now we are in December 2023 and arguably all the information to add 2023 to the current analysis could be done. It is true that this may somewhat complicate the message, as 2023 may show to be even more extreme than 2022 in some areas, but also maybe not. I would urge the authors to consider to add 2023 also to the analysis (at least in Figure 9 and maybe 10) to ensure their analysis does not become a bit obsolete before it is even published. Additionally, I would ensure something is said about 2023 in the discussion/conclusion.

The reviewer is right in the points raised here. Adding information on 2023 is indeed informative but also complicates the message, as pointed out. We included information on 2023 but do not treat it with any particular highlight, since overall, from the LST perspective, 2022 is still exceptional even when 2023 is considered. Also from the 2m temperature perspective, and according to the information provided by the Copernicus Climate Change Service, summer 2023 *over Europe* ranks as the fifth warmest year on record (https://climate.copernicus.eu/summer-2023-hottest-record).

As suggested, the following changes were made:

- 2023 was included in the (new) Figure 9 (Time series of the percentage of land area affected by M_d>2 (...)). Please see the new figures in the bottom of the document. L421 now reads"(...) together with the corresponding data from the individual years from 2004 to 2023 (...)". Since there is not much to say about that particular year, no further changes in the text associated to this figure were introduced.
- 2023 was included in the new Figure 10 (Ranking of summers over the study period (...)". L429 now reads: "In Figure 109, all the summers in the LST data record (2004-2023) are ranked (...)". Again no further considerations were made given the low positions in the different rankings for 2023.
- 2023 was included in the new Fig 11 (Year where the record maximum average LSTMax occurred (...)). Here a few changes were made in the overal description. L453-464 now read "In the JJA anomaly, new maxima were set for large areas of Northern Iberia, France, Southern Germany, Italy and Hungary (in bright cyan). These areas have strong signals over the individual monthly maps as well. The 2018 (dark

grey) heatwave still holds the record for large parts of North Central Europe (Hoy et al., 2020), while the 2010 (red) heatwave set the overall record over Russia (Barriopedro et al., 2011). In the June map, the 2019 (light grey) heatwave introduced records for this month over a large part of North Central Europe, while 2023 (light blue) set new records for Northern France and the Benelux area. In July, 2022 (bright cyan) set new monthly records over South Central Europe, 2006 (orange) set the July record for large areas from France to the Baltic countries, while the in the north Central Europe the record was set by the 2018 (dark grey) heatwave. In August the year 2022 set new records for large areas from northwest Portugal, to France and the British Isles up to the Baltic countries. The 2015 (light brown) heatwave still holds the record for August over areas such as Poland, Belarus and west Ukraine. It is worth noting that 2003 set JJA temperature records over large areas of Western Europe (Sousa et al., 2020; Lhotka et al., 2018), but since that year is not within the period covered by the LSA-SAF LST dataset, it does not show up in 11."

- Sentence was added to the conclusions L482-484"This study also showed that despite some areas in June and August had their highest monthly anomalies in 2023, the 2022 summer as a whole remains as exceptionally warm, ranking in first in all considered metrics."

While LST_max is seen to have differences with T2M_max and SKT_max, which is interesting and informative, I think this analysis could be pushed a bit more to give a bit more insight of these difference. Specifically these seems to scale with magnitude, and it would be wise to try to qualtify/visualize this to provide some info on the non-linearity of the relationship. I would thus strongly recommend some graph maybe showing the deltas of all pixels on the y-axis and the magnitude of each pixel on the x-axis (for both T2M and SKT separately)

Indeed the suggested analysis was missing, thank you for asking for it. The proposed plot is provided below (please, see end of document).

The following text was added to the end of section 3.2, together with the new figure L324-335: "In Figure 5, the temperature differences are further analyzed as a function of the absolute LST_Max. Their behaviour is somewhat consistent across the absolute LST_Max range. For instance, for lower LST_Max, both differences are small and negative. A large part of this can be explained by persistent clouds, which if undetected, could introduce a negative bias in LST (Martins and Dutra, 2022; Trigo et al., 2021; Martins et al., 2019). These situations are however relatively infrequent. For mid-range LST_Max, differences are generally positive, with larger LST_Max-T2m-Max, especially in July when they reach around 2°C. For LST_Max around 45-55 °C, temperature differences are relatively lower, but they increase again for very high LST_Max".

On another take, I was quite interested in knowing more about the specific performance and appropriateness of the all-sky LST (versus the clear-sky LST). It seems that it is a bit taken from granted that it is assumed to be better (because more gap-filled). However, this paper could be the nice opportunity to evaluate better how it performs against the clear-sky in terms of relationship with other indices (SKT_max, T2M_max), and thereby giving an extra relevance for this paper.

Indeed a paper focusing on an extended validation of the product is missing (it is in preparation by the team). But it is not exactly true that no validation has been performed. A feasibility study by Martins et al (2019) already performed a validation of a preliminary version of this product against 3 in situ stations specifically designed for LST validation and compared to another all-sky LST product from AMSR-E. Furthermore, a validation report was approved internally by EUMETSAT after careful external review (a pre-requisite for product release), where an exhaustive comparison against 33 stations from several land monitoring networks and to ERA5-Land skin temperatures. This report is available at the LSA-SAF website and is already cited in the manuscript. Again, we decided to go forward with this manuscript after formal release of the product, but without waiting for the publication of the validation paper, due to the exceptionality of the event and because we assumed the validation report already provided enough information to the interested reader. Moreover, the dataset used for the cloudy-sky SKT derived from a surface energy balance scheme has been published (Barrios et al. 2024, https://doi.org/https://doi.org/10.1002/gdj3.235). This reference has been added, see below.

From the presentation point of view, figures would be clearer and more accurate using an appropriate geographical projection for Europe which respects the concept of equal area, thereby showing better the extent of the meteorological event described in this paper. I would recommend to use the Inspire LAEA for Europe.

Thank you for your suggestion. Figures were adapted to use the suggested projection.

When answering to this comment, we decided to slightly change the domain (now limiting the upper limit to 60N). We have also applied tighter quality control when computing the monthly means. Both of these modifications led to slight differences in the reported values and even in the relative position of each year in the rankings of Fig 9 (Fig 10 in the new manuscript). In section 2.5 this is now explained L194-195: "When computing monthly means, it was ensured that at least 85% of the days in each month were available. This prevents spurious values in the disk edge to contaminate the monthly value."

**Specific comments...**

- L60 : "led to twice the yield", do you mean increase in yield?

Sorry if this was unclear. The sentence was rephrased L58-60: "The combined effects of drought and extreme heat also led to a wide range of economic impacts, namely an overall crop loss (particularly cereal) of 9% with respect to the previous years' five-year average production (FAO, 2022), causing a generalized increase in food and grocery prices"

- L72: yes, but this does not relate necessarily to NRT in general

Not sure what you mean here. EUMETSAT is indeed making efforts to increase the usage of their datasets on climate applications (e.g., https://www.eumetsat.int/what-we-monitor/climate, https://www.eumetsat.int/what-we-do/monitoring-climate, https://www.eumetsat.int/climate-data-records, last visited in 7 January 2024)

- L86: perhaps it would be welcome here to add briefly a bit more on how this all-sky is produced, or rather, how come this is not the standard? what assumptions are being made to estimate what is below the clouds? I know this is explained later, but a word of two talking about energy balance would fit well

Indeed. Added L99-102: LSTs under cloudy conditions are estimated using a surface energy balance scheme (Barrios et al., 2024), which is used at the LSA-SAF to estimate evapotranspiration and surface turbulent fluxes along with the cloud-sky LST. The scheme uses radiation flux and vegetation products from the LSA-SAF, H-SAF soil moisture and a few screen-level variables from ECMWF as main inputs."

- L101: perhaps good to mention how many MSG satellites there are

Indeed, L120 corrected to "onboard the four Meteosat Second Generation Satellites"

- L114: what does overall accuracy of 0K mean here exactly? it reads as if there is no error whatsoever. is it not rather that you mean there is no bias?

We are using the terminology proposed by the Committee on Earth Observation Satellites Working Group on Calibration and Validation Land Product Validation Subgroup in the Land Surface Temperature Product Validation Best Practice Protocol (Guillevic et al., 2018). We would rather stick to that terminology. However, we added L133 "(bias)" in the text as suggested, for increased clarity.

- L117: which meteo variables from ERA5 are needed?

Information is included as follows L129-130 "and screen-level meteorological variables (including 2 m temperatures, 2 m dewpoint temperature, 10 m winds and surface pressures) from the European Centre for Medium-Range Weather Forecasts (ECMWF).

- L169: please make it explicit here that this is how you define "hot days" in this paper. It did not seem so clear to me later on that the definitin was here

Rephrased to L200 "These days where P90 is exceeded are hereafter referred to as *hot days*."

- Figure 2: in panel a, please change the colourscale as this one is not colourblind friendly

We selected a new colorscale, hoping the problem is now mitigated. Please see new set of figures in the end of this document.

- L269: I suppose the fact LAI is prescribed should also be mentioned here

Corrected to L297 "physiographic fields such as vegetation cover/LAI (which are static)"

- Figure 4: would be nice to relate this with absolute values of LST_max/T2M_max. Are the deviations larger particularly where the temperatures are larger? it would surely seem so.

We hope that the answer to the 2ⁿᵈ major point answers to this.

- Figure 7: the colourscale on the right column of figures is als onot ideeal for colourblind people as it goes from green to red.

The new figure version uses the same colorbar for both columns

- Figure 8: graphically it is difficult to appreaciate how the corresponding areas are for non-2022 years, as these are not full. I wonder if there would not be a way to make things more comparable. Maybe shading also (with different colours maybe) the 3 next years with the largest cumulated max Area under $Md(Td)>2$.

The idea of this plot is not actually to focus on areas under curves. The reviewer' suggestion made the plot confuse, because there are many years under comparison. We prefer to leave the plot as it was (but including information for 2023). We acknowledge that the comparison of the areas under the curves are indeed relevant, which is why they are directly compared in the ranking figure. The idea here is to focus on the time dimension, by following the evolution of the proportion of land area under extreme heat conditions (which are defined here as pixels with $Md>2$.

- Figure 9: it would really be interesting to add 2023 on this graph

Done.

**New plots:**

[Figure]

Figure 1 Panels representing the anomalies of the synoptic atmospheric configuration over Europe for two seasons in 2022, as given by ERA5: MAM (panels (a) and (c)) and JJA (panels (b) and (d)). Panels (a) and (c) show $T_{850}$ anomalies (in colour), and $Z_{500}$ (black contours). Dotted areas denote areas where $T_{850}$ was above its 95th percentile. Panels (b) and (d) show the normalized anomaly of accumulated precipitation (in colour) and $\vec{v}$ anomalies (black arrows). Dotted areas denote areas where $\boldsymbol{precip}$ anomaly was below the 10th percentile. All anomalies were computed with respect to the 1981-2022 reference period. Arrows are spaced 2°x2° for the sake of readability.

Fig 2

[Figure]

Figure 2 - (a) JJA median of $LST_{Max}$ for the period 2004-2021, while (b, c, d, e) are seasonal $LST_{Max}$ anomalies for 2012, 2018, 2019 and 2022.

Fig 3

[Figure]

Figure 3 - - $LST_{Max}$ monthly anomalies for (a) June, (b) July and (c) August 2022 over Europe.

Fig 4

[Figure]

Figure 4 - Comparison between $LST_{Max}$ monthly anomalies and the corresponding anomalies using reanalysis $SKT$ *(left) and $T2m$ data (right). Comparisons are made for June (a, b), July (c, d) and August (e, f).*

Fig 5

[Figure]

*Figure 5 - Mean differences between $LST_{Max}$ and $SKT_{Max}$ (orange, diamonds) anomalies and between $LST_{Max}$ and $T2m_{Max}$ (blue, circles) anomalies as a function of mean $LST_{Max}$, for June (left panel), July (center panel) and August (right panel). On top, the number of pixels used in the calculation. Whiskers represent the standard deviation over each interval.*

Fig 6

[Figure]

*Figure 6- (a) Number of JJA hot days detected using the $LST_{Max}$ (i.e., days when $LST_{Max} > P_{90}$). (b) Total JJA HMWI derived with $LST_{Max}$. (c, e) Differences between the number of Hot Days obtained with $LST_{Max}$ and with $SKT$ and $T2m$, respectively. (d, f) Difference to the $SKT$-based HWMI and T2m-based HWMI, respectively. The blue square in (a) denotes the area used for the extraction of timeseries data which are analysed below.*

**Fig7 (same as previous fig6)**

[Figure]

[Figure]

[Figure]

*Figure 7- (top) Evolution of $LST_{Max}$ (green curve) and the respective $P_{90}$ (dashed curve). Hot days are marked as a yellow circle at the top; if they belong to a heatwave (set of 3 or more consecutive days), they are marked as a red circle. (middle) Explicit differences between $LST_{Max}$ and the $P_{90}$. (bottom) Daily heatwave magnitude, $M_d$ is in blue and the accumulated HWMI is in red, with values in the right axis. All data are area averages from the blue box in Figure 6.*

Fig 8

[Figure]

*Figure 8- (left) **FVC** monthly anomalies and (right) SWI, for June (top), July (middle) and August (bottom). Reference period is 2004-2021.*

Fig 9

[Figure]

Figure 9 - Time series of the percentage of land area affected by $M_d > 2$, from June 1st to August 31st. The red bolder curve represents 2022 data, while other colors represent the same variable for the other years in the data record. Stars mark the days where the area where $M_d > 2$ in 2022 was the greatest over all years.

Fig 10

[Figure]

Figure 10 - Ranking of summers over the study period according to (left) their mean $LST_{Max}$ anomaly, (middle) the average fraction of area covered by extreme heat conditions ($M_d > 2$) and (right) area-averaged HWMI. Colours are mainly for illustrative purpose, where each year was classified according to the severity associated to each parameter (from less severe in blue to extremely severe in dark red).

Fig 11

[Figure]

*Figure 11 - Year where the record maximum average $LST_{Max}$ occurred for the periods (a) JJA (b) June (c) July and (d) August.*

---

## Author Comment (AC2)

Review of: "A satellite view of the exceptionally warm summer of 2022 over Europe" submitted to NHESS by Martins et al.

In this study the authors perform a satellite-based analysis of the unusually high temperatures experienced during the summer of 2022 in Europe. They use a wide array of satellite products from MSG-SEVIRI, including the LSA-SAF All-Sky LST product. The manuscript is well-written, the methodology is clearly outlined, and the conclusions are well-supported by the results. The chosen topic also aligns with the scope of the "Natural Hazards and Earth System Sciences" journal. Although the study may not introduce particularly novel insights, remotely-sensed LST is not used so frequently for monitoring heatwaves. Consequently, this manuscript contributes valuable information to the existing body of knowledge.

The authors would like to thank the reviewer for the generally positive feedback, all the  valuable suggestions and for the time spent on carefully reviewing this manuscript. We address each of the recommendations below.

However, prior to publication, I recommend that the authors make substantial improvements in two aspects of their study: (a) establishing better connection between their methodology and findings with previous works in the literature, and (b) enhancing the robustness of their methodology and conclusions when comparing LST with reanalysis variables.

I list below my concerns in more detail, along with some additional minor issues.

Major issues:

1) As the authors note in the abstract that the climate applications of LST are still poorly explored, it is crucial for them to offer a more in-depth analysis of previous studies, elucidating also points of agreement or disagreement in the findings. Therefore, additional discussion is necessary in the Introduction regarding past research that has utilized remotely sensed LST for studying heatwaves, and this discussion should also be incorporated when presenting the findings of the current study. While some relevant works have already been cited in the references, they would benefit from more comprehensive discussion. Furthermore, there is room to expand the list of prior works that contribute to the overall understanding of the topic.

We agree that not much context is given with regards to previous works using LST for the study of heat waves. But the truth is that there are really not many high impact studies on that topic. Reiners et al. (2023) performed a meta-analysis on the different topics regarding LST in the literature, and showed that LST datasets have not been much used to study heatwaves. According to the authors, this is due to 1) the lack of long time series (i.e., longer than 30 years), 2) the absence of reliable all-sky LST datasets, 3) the lack of intercomparison between different LST timeseries and intercomparison among different climate indicators

and finally, 4) the lack of validation over heterogeneous sites. Although the meta-analysis performed by the authors is not extremely exhaustive, we agree with their reasoning and conclusions.
Nevertheless it is true that a few recent studies have used LST to study heatwaves and brought important insight into the topic, in particular the studies by Good et al (2022) and Agathangelidis et al. (2022).
Several adaptations on the introduction have been introduced in order to reflect that insight but also to explain why we think that using the LSA-SAF All Sky LST product may help to overcome some of the limitations of those studies.
L79-84 now reads " Despite the relatively good temporal and spatial coverage of Land Surface Temperature (LST) products over areas with significant heatwaves in the past decades (such as south-central Europe), Reiners et al. (2023) showed that these products have not been much used to study these phenomena. According to these authors, this is due to 1) the lack of long time series (i.e., longer than 30 years), 2) the absence of reliable all-sky LST datasets, 3) the lack of intercomparison between different LST timeseries and intercomparison among different climate indicators and finally, 4) the lack of validation over heterogeneous sites."
L84-91 now reads "Good et al. (2022) also demonstrated the usefulness of LST for climate monitoring and found good agreement between LST anomalies derived from the ESA Climate Change Initiative LST (Pérez-Planells et al., 2023) datasets and in situ 2-meter temperature anomalies derived from the ECAD/E-OBS dataset (Cornes et al., 2018). Nevertheless, the association between LST and 2-meter temperature anomalies under heatwave conditions is not always straightforward (Agathangelidis et al., 2022; Mildrexler et al., 2011), as they can differ substantially both in spatial and temporal extents, especially towards higher temperatures. An additional limitation on the analysis of LST daily timeseries based on polar orbiting satellites (Agathangelidis et al., 2022; Good et al., 2022) is that measurements over a particular area are obtained at slightly different times of day and with different viewing and illumination geometries."

2) Regarding the comparison with reanalysis I have the following remarks:

Why did the authors choose to use T2m from ERA5 instead of ERA5-Land? It appears to me that the latter would be more suitable for consistency, considering that SKT comes from the ERA5-Land product.

In the initial submission we have chosen to use ERA5 T2m as reference dataset since it has observational constraints due to the land data assimilation in ERA5 which is not present in ERA5-Land. To be more precise, ERA5 merges the model T2m diagnostic with in-situ observations via an optimal interpolation scheme, while T2m in ERA5-Land is only a result of the land surface model diagnostic.
We acknowledge that T2m from ERA5-Land is more consistent with ERA5-Land SKT and have updated all the manuscript accordingly.  This change did not impact the overall study conclusions. In particular, in section 2.4 this choice is explained: "Although ERA5 is by far the most widely used dataset in heatwave studies, in this study  estimates from ERA5-Land are used, as this choice allows to focus on the physical differences between T2m the SKT ,

without having to consider differences introduced by comparing different modelling systems and spatial resolutions."
Please see the new set of figures in the end of this document.

Lines 262, 302, 370: The authors in various parts of the text highlight the difference in physical meaning for T2m, however the patterns of T2m in Figure 4 (and also throughout the manuscript) appear to closely resemble those of SKT. Have the authors examined the difference between this pair of variables, similar to what was done with LST? It may be beneficial to include these figures as well.

The reviewer is right. That comparison is now included in figure 4.
Also in section 3.2 L318-322 now reads: "Regarding SKT_Max-T2m_Max, one should note that these are entirely produced by reanalysis alone. The comparison reveals that thermal contrasts between SKT_Max and T2m_Max are much smoother than those between LST_Max and T2m_Max. Since the surface sensible heat flux is proportional to this difference, this suggests that sensible heat fluxes are weaker in ERA5-Land under extreme heat conditions when compared to observations (i.e., LST-T2m differences), although other model parameters might play a role in the sensible flux modulation (e.g. surface roughness)."

Line 273: Please provide a quantitative measure of the differences in temperature anomalies over burned areas between LST and SKT.

A dedicated study of the temperature anomalies over burned areas is out of the scope of this manuscript. A qualitatively rigorous analysis would require using a burned areas dataset, which is not guaranteed to be consistent with the datasets used here (i.e., it would not be straightforward to assign burned pixels to their respective thermal anomalies and then perform a quantitative analysis as requested). The reference to burned area pixels is intended to be illustrative. Nevertheless, a rough estimate is provided, and the sentence was edited as follows (L302-307): "For instance, temperature anomalies over burned areas are generally higher when they are determined based on observations, than when they are based on the model , since the relevant changes associated with burn areas in the physiographic fields is not included in ERA5 (e.g., surface emissivity, albedo, vegetation cover, etc.). Some of these fire scars are visible in Figure 3 (e.g., over the Iberian Peninsula). Close inspection of pixels roughly corresponding to burned areas associated to fires occurred in July 2022, namely in northwest Spain (Castilla and Leon) and in the south (Andalusia), reveals LST-SKT mean differences of up to 14°C in the August maps."

Lines 275 – 288: Please present aggregated statistics about the magnitude of the difference between LST, SKT, and T2m anomalies.
We added a new figure 5 to answer this point from the reviewer (in a combination with remarks made by the other reviewer). In th end of section 3.2 one may now read:
In Figure 5, the differences are further analyzed as a function of the absolute LST_Max . Their behaviour is consistent across the absolute LST_Max range. For instance, for lower LST_Max, both differences are small and negative. A large part of this can be explained by persistent clouds, which if undetected, could introduce a negative bias in LST (Martins and Dutra, 2022; Trigo et al., 2021; Martins et al., 2019). These situations are relatively

infrequent. For mid-range LST_Max, differences are generally positive, with larger LST_Max-T2m_Max especially in July when they reach around 2°C. For LST_Max around 45-55 °C, temperature differences are relatively lower, but they increase again for very high LST_Max.

[Figure]

Figure 5 – Mean differences between and (orange, diamonds) anomalies and between and (blue, circles) anomalies as a function of mean , for June (left panel), July (center panel) and August (right panel). On top, the number of pixels used in the calculation. Whiskers represent the standard deviation over each interval.

Therefore, despite the good correlations between LST and T2m found by Good et al. (2022), these results show that there is a wide range of situations where these temperatures may be very different."

Line 335: I may have missed it, but has SWI been defined? In addition, is SWI derived using inputs from MSG that are common with those used to derive the LST All-Sky product? If yes, this could explain part of the strong resemblance between the two, and it should be mentioned in the conclusions.

The reviewer is right, sorry for that. We added in section 2.2 L161-164: "In the case of the selected dataset, soil moisture is linearly rescaled between wilting point (0) and field capacity (1), defining the Soil Wetness Index (*SWI*). In this work, an SWI average of the first three layers (i.e., down to 1 m below the surface) from the daily data is used to compute monthly means, from which the 2004-2021 climatology and anomalies are derived."

SWI computed from the HSAF soil moisture is based on scatterometer data, so it is independent from MSG. But then it is indeed used as input, only to the cloudy part retrieval of the All-Sky product. As the reviewer said above, heatwaves mainly occur over prolonged periods of clear skies as the excess solar radiation is one of their key ingredients. However, it is true that despite of that, soil moisture and All-Sky LST are not fully independent.

In section 3.4 L385-388, the following text was added: "*SWI* is used as input to the surface energy balance model that is used to derive cloudy sky *LST*. However, it can be inferred that most of the retrievals under heatwave periods are made for clear sky. Therefore, it can be assumed that LST, FVC and SWI are mostly independent from each other."

In the conclusions L494-497: "It should be noted however that *SWI* is used as input to the cloudy sky *LST* retrieval. However, most of the retrievals under heatwave conditions are made under clear skies, so most of the *LST* signal is coming from the infrared retrieval, and not from the surface energy balance scheme used for cloudy sky retrievals."

Minor issues / Technical comments

Lines 58 – 63: This discussion seems somewhat beyond the manuscript's scope. Could you rephrase it to be more concise?

Rephrased to L58-60: "The combined effects of drought and extreme heat also led to a wide range of economic impacts, namely an overall crop loss (particularly cereal) of 9% with respect to the previous years' five-year average production (FAO, 2022), causing a generalized increase in food and grocery prices."

Line 84: Why is the significance of an all-sky product for heat extremes emphasized, considering that clear-sky conditions are typically the norm?

Cloudy-sky scenes have been acknowledged as one of the major factors hampering the use of satellite LST in climate studies (e.g., Gouveia et al, 2022; Reiners et al., 2023). The reasons for it are already explained in that sentence and are mainly related to the presence of gaps in the timeseries. The following sentence is added to further detail the explanation L94-96: "In particular, these discontinuities hamper a correct count of the number of hot days (especially the consecutive hot days, which are relevant for the determination of heatwave conditions), or a correct assessment of the spatial extent of extreme heat conditions." and a few sentence later L104-107 "By using a product that fills in those gaps using a physically-based algorithm (i.e., estimates a land surface temperature value taking into account the changes in radiative fluxes under clouds, as well as vegetation state and soil moisture conditions), interpolations that are many times unphysical are avoided. Although clear-sky conditions are typically the norm for heatwave conditions, clouds are nonetheless frequent and ubiquitous."

Previous analysis (not shown) revealed a significant decrease in the above mentioned indicators (e.g number of hot days) when the clear sky only LST product is used (when compared to ERA5 or the All-Sky LST-based indicator), especially over cloud prone areas.

Line 114: It is unclear what the authors mean with "overall accuracy of 0.0 K"? Maybe bias is a more suited word?

We are using the terminology proposed by the Committee on Earth Observation Satellites Working Group on Calibration and Validation Land Product Validation Subgroup in the Land Surface Temperature Product Validation Best Practice Protocol (Guillevic et al., 2018). We would rather stick to that terminology. However, we added "(bias)" in the text (L133) as suggested, for increased clarity.

Line 121: Have cases of active fire been included in the analysis? If yes, can the authors provide justification how this inclusion may not significantly impact the results about heatwaves?

No special treatment is applied in the case of active fires in this product version. They are indeed sources of uncertainty at a very local level. Reasons include: 1) fire temperatures

saturate the split-window channels (so the IR retrieval is not properly calibrated to deal with them, nor the used channels have the required sensitivity), 2) they produce smoke clouds / pyrocumulus that may be challenging for the cloud mask algorithm, 3) for cloudy retrievals, the surface energy balance scheme is also not designed to resolve the surface energy balance under such extreme conditions. However, we don't expect that the results presented here are particularly affected by active fires because 1) they often occur over areas smaller than the SEVIRI pixel size and 2) all statistics presented here are temporal or spatial averages, so the effect of active fires was averaged out.

Line 142: From this point onward, it appears that the section numbering is disrupted.

Corrected, thank you.

Lines 165 – 175: I recommend presenting the description of monthly and seasonal anomalies first, followed by the heatwave definition and metrics. This also aligns with the presentation in the results section.

We agree. The anomaly estimation procedure was moved to the beginning of the section L191-195 (and slightly reworded accordingly), and the heatwave definition and metrics are presented afterwards.

Figure 2, 5: For clarity, I suggest avoiding the use of a dash (-) as a separator in subplot titles. This is particularly important to prevent confusion, as the minus sign is also used in titles to indicate differences.

We agree with this suggestion. Dashes were removed from plot titles.

Line 260: I believe you meant to write "ERA5-Land STK and ERA5 T2m"

Correct. But given we now use ERA5-Land for T2m, the sentence should be kept as it was.

Lines 351-352: Is there a reference to support this claim?

We are not sure we understood this point. The studies by Johannsen et al., 2019; Nogueira et al., 2020, 2021; Duveiller et al., 2022, which are cited a couple of sentences later, focused precisely on the misrepresentation of skin temperature due to inconsistencies in the physiographic fields in H-TESSEL (the ERA5 and ERA5-Land surface scheme), particularly those associated to vegetation state.

Line 352: Is it ERA5 or ERA5-Land? It seems the authors use these terms interchangeably also in other parts of the text.

Corrected to ERA5-Land as suggested.

Line 440: It is not clear to me what is meant by "for satellite observations these errors are mostly mitigated". While satellite-based observations have their own sources of

uncertainty, these do not typically involve "interpolation" or "model" errors, as suggested here.

Indeed the sentence was unclear, thank you for pointing this out. It was rephrased to L488-490: "For satellite observations, spatial coverage is much higher, and therefore spatial interpolation errors are mostly absent (although there are relatively small uncertainties associated to geolocation and regridding from the original satellite observations to regular grids such as those used in this study)."

**New plots:**

[Figure]

Figure 1 Panels representing the anomalies of the synoptic atmospheric configuration over Europe for two seasons in 2022, as given by ERA5: MAM (panels (a) and (c)) and JJA (panels (b) and (d)). Panels (a) and (c) show $T_{850}$ anomalies (in colour), and $Z_{500}$ (black contours). Dotted areas denote areas where $T_{850}$ was above its 95th percentile. Panels (b) and (d) show the normalized anomaly of accumulated precipitation (in colour) and $\vec{v}$ anomalies (black arrows). Dotted areas denote areas where **precip** anomaly was below the 10th percentile. All anomalies were computed with respect to the 1981-2022 reference period. Arrows are spaced 2°x2° for the sake of readability.

Fig 2

[Figure]

Figure 2 - (a) JJA median of $LST_{Max}$ for the period 2004-2021, while (b, c, d, e) are seasonal $LST_{Max}$ anomalies for 2012, 2018, 2019 and 2022.

Fig 3

[Figure]

Figure 3 - - $LST_{Max}$ *monthly anomalies for (a) June, (b) July and (c) August 2022 over Europe.*

Fig 4

[Figure]

*Figure 4 - Comparison between $LST_{Max}$ monthly anomalies and the corresponding anomalies using reanalysis **SKT** (left) and **T2m** data (right). Comparisons are made for June (a, b), July (c, d) and August (e, f).*

Fig 5

[Figure]

*Figure 5 - Mean differences between $LST_{Max}$ and $SKT_{Max}$ (orange, diamonds) anomalies and between $LST_{Max}$ and $T2m_{Max}$ (blue, circles) anomalies as a function of mean $LST_{Max}$, for June (left panel), July (center panel) and August (right panel). On top, the number of pixels used in the calculation. Whiskers represent the standard deviation over each interval.*

Fig 6

[Figure]

Figure 6- (a) Number of JJA hot days detected using the $LST_{Max}$ (i.e., days when $LST_{Max} > P_{90}$). (b) Total JJA HMWI derived with $LST_{Max}$. (c, e) Differences between the number of Hot Days obtained with $LST_{Max}$ and with $SKT$ and $T2m$, respectively. (d, f) Difference to the $SKT$-based HWMI and T2m-based HWMI, respectively. The blue square in (a) denotes the area used for the extraction of timeseries data which are analysed below.

**Fig7 (same as previous fig6)**

[Figure]

[Figure]

[Figure]

*Figure 7- (top) Evolution of $LST_{Max}$ (green curve) and the respective $P_{90}$ (dashed curve). Hot days are marked as a yellow circle at the top; if they belong to a heatwave (set of 3 or more consecutive days), they are marked as a red circle. (middle) Explicit differences between $LST_{Max}$ and the $P_{90}$. (bottom) Daily heatwave magnitude, $M_d$ is in blue and the accumulated HWMI is in red, with values in the right axis. All data are area averages from the blue box in Figure 6.*

Fig 8

[Figure]

*Figure 8- (left) **FVC** monthly anomalies and (right) SWI, for June (top), July (middle) and August (bottom). Reference period is 2004-2021.*

Fig 9

[Figure]

*Figure 9 - Time series of the percentage of land area affected by $M_d > 2$, from June 1st to August 31st. The red bolder curve represents 2022 data, while other colors represent the same variable for the other years in the data record. Stars mark the days where the area where $M_d > 2$ in 2022 was the greatest over all years.*

[Figure]

Fig 10

[Figure]

*Figure 10 - Ranking of summers over the study period according to (left) their mean $LST_{Max}$ anomaly, (middle) the average fraction of area covered by extreme heat conditions ($M_d > 2$) and (right) area-averaged HWMI. Colours are mainly for illustrative purpose, where each year was classified according to the severity associated to each parameter (from less severe in blue to extremely severe in dark red).*

Fig 11

[Figure]

*Figure 11 - Year where the record maximum average $LST_{Max}$ occurred for the periods (a) JJA (b) June (c) July and (d) August.*